# Quantitative comparison of three-dimensional bodies using geometrical properties to validate the dissimilarity of a standard collection of 3D geomodels

Friedrich Carl[1], Jian Yang[2], Marlise Colling Cassel[3], Florian Wellmann[2], Peter Achtziger-Zupančič[4]

[1]Chair of Engineering Geology and Hydrogeology, RWTH Aachen University, Lochnerstr. 4-20, 52064 Aachen, Germany
[2]Chair of Computational Geoscience, Geothermics and Reservoir Geophysics, RWTH Aachen University, Mathieustr. 30, 52074 Aachen, Germany
[3]Chair of Geology and Sedimentary Systems and Geological Institute, RWTH Aachen University, Wüllnerstr. 2, 52062 Aachen, Germany
[4]Fraunhofer Research Institution for Energy Infrastructures and Geotechnologies IEG, Aureliusstr. 2, 52062 Aachen, Germany

*Correspondence to*: Friedrich Carl (carl@lih.rwth-aachen.de)

**Abstract.** The quantification of 3D structural shapes is a central goal across multiple scientific disciplines, serving purposes such as image analysis and the precise geometric characterization of objects. This study proposes a methodology for the shape quantification based on a set of geometrical parameters in 2D sections of 3D geological shapes and establishes a set of synthetic regular geometries as benchmark models in 3D geomodeling approaches. The proposed methodology is demonstrated on a number of simple geometric bodies and the benchmark models to assess their geometrical dis-/similarity. The dimensions of the structures are measured perpendicular and vertically to their horizontal main axes on a fixed amount of cross sections. Furthermore, gradient and curvature measurements on these cross sections are conducted. A subsequent multi-step data analysis provides insight into the main geometrical characteristics of the structures and visualizes differences between various datasets: Analysis of extension measurements reveals the anisotropy of structures, the existence of overhangs and the character of the top surface of an investigated structure. Analyzing the gradients and curvatures offers information on the slopes of the lateral walls of the structure and its sphericity as well as top surface. Kullback-Leibler divergence is utilized to quantitatively compare individual parameter distributions. Dimensionally reduced cluster analysis groups and systematizes input structures based on the combined statistical parameters and serves for the identification of benchmark models showing large geometrical similarity. It is expected that the methodology and set of benchmark models will aid in advances to model, analyse and compare subsurface structures based on sparse data, as our framework can be used for an initial structural approximation prior to modeling, for the setup of the interpolation method and for the falsification of probabilistic model realizations after interpolation.

# 1 Introduction

## 1.1 Shape quantification and comparison – previous studies and gaps in current research

The quantitative comparison of three-dimensional (3D) objects plays a crucial role in various scientific fields, including geology, computer science and engineering (see e.g. Cardone et al., 2003; Celenk, 1995; Wellmann and Caumon, 2018). Shape quantification aims at the numerical characterization of the geometry of objects, with their dis-/similarity not solely being a mathematical metric but also being dependent on the specific context (Laga et al., 2019). Accurate shape quantification independent of the objects' orientation is essential for applications such as geological modeling, resource management and structural analysis, where understanding the geometric properties of objects can inform decision making and enhance predictive capabilities.

Shape quantification can be complex when dealing with static 2D images of 3D bodies (see Laga et al., 2019), but when rotatable objects in 2D or 3D are available, basic geometrical parameters can be applied. This is commonly proposed in material science, where studies focus on sand grain analysis. In these studies, the range of shape parameters in 2D and 3D include (but are not restricted to) principal dimensions, volumes, aspect ratios, radii, sphericities, convexity, circularity, roundness and compactness (Altuhafi et al., 2013; Cox and Budhu, 2008; Zhao and Wang, 2016), that partially describe similar structural characteristics. Furthermore, the shapes of aggregate particles in building materials have been analyzed using parameters like sphericity, angularity, aspect ratios, gradients and radius indices (Al-Rousan et al., 2007), and volcanic cinders have been assessed looking at elongation, roundness, and roughness (Nie et al., 2023). Similar analytical approaches can be valuable to study the geometry of subsurface structures, although at much larger scales and a higher structural complexity: The shape of individual geo-bodies can be of interest for resource exploration and storage of materials like for instance nuclear waste. However, geoscientific studies applying similar parameters as used in the mentioned material-scientific studies are rare: Gardoll et al. (2000), for instance, determine the aspect ratio, blockiness, elongation, compactness, complexity, roundness, spreadness and squareness of geological bodies from map data to assess the exploration potential for orogenic ore deposits. This is a highly specialized application though, usable for shallow horizontal data, but being inapplicable to (sub-)vertical input data. Instead of relying on geometrical parameters for the shape quantification of a single geo-object, in geosciences advances for the shape comparison of structural models are more common. These are mainly related to uncertainty assessment and quantification within geological models and often approached with distance metrics. For instance, Schweizer et al. (2017) apply the Jaccard distance and the normalized city-block distance as measures for model dissimilarity, while Suzuki et al. (2008) propose the usage of the Hausdorff distance for the same purpose. In contrast, Lindsay et al. (2013) developed an approach for model comparison not relying on such distance metrics: To determine the similarities between 101 realizations of a 3D composite geomodel based on the same perturbed input dataset, a set of geometrical "geo-diversity" parameters (e.g. formation depth, volume, contact surface curvature) are calculated on all stratigraphic units. The resulting datasets are analyzed in their ranges to determine endmember model realizations. Furthermore, principal component analysis is employed to determine which geometrical characteristics contribute most to spatial uncertainty and to detect

realization outliers for the combined geodiversity metrics. Despite the lack of geoscientific studies approaching shape quantification with simple geometrical parameters, the necessity of basic shape assessment in geomodeling is recognized as most commonly used geomodeling software are capable of obtaining simple geometrical properties like surface areas, aspect ratios and volumes from modeled 3D elements. However, these functionalities are error-prone determining basic geometrical properties in varying directions, like the extent along the horizontal main axes of a given irregular structure – a property of interest for the exploration of geo-bodies for storage purposes. An example of such an application can be found in the storage of high-level nuclear waste: In Germany, currently, intrusive salt bodies with varying internal structures as well as crystalline intrusives potentially exhibiting lateral zonation are considered as potential storage sites (BGE, 2020).

In addition to these limitations in the analysis of 3D geo-bodies, geometrical characteristics of structures are hard to quantify prior to geomodelling as well, when input data is most commonly available in 1D (i.e. boreholes) and/or 2D (e.g. seismic sections). At this early stage within a modeling workflow, conceptual models are established based on sparse data, local geological knowledge like the regional geological history and universal geological knowledge such as common laws and principles (Parquer et al., 2025) but also defined spatial factors known to be related to certain variables of interest like resources (Gardoll et al., 2000). The identification of important geometrical features and the establishment or selection of an appropriate conceptual model can have a considerable impact on how realistic/reasonable model realizations are, thus influencing decision-making and the accuracy of predictions (Bond et al., 2007). Therefore, approaches to geometrically quantify available input data and to compare datasets to established conceptual models are valuable.

Given this identified current lack of analytical capabilities for the geometrical assessment of both unmodelled input data as well as modeled structures whose evaluation shall be direction-dependent, this study proposes a novel methodology for the quantitative description, comparison and systematization of datasets using a set of geometrical parameters. While the method development will be visualized based on explicitly modelled 3D geometries, it can be applied to lower dimensional data as well. In the present study, the algorithm is applied to a set of 36 3D geometries approximating subsurface structures of varying rock types, intended to act as benchmark models in geomodeling approaches. By demonstrating the quantification algorithm on these 3D bodies called "standard geometries", their geometrical dis-/similarity is analyzed. Furthermore, the methodology has been applied to a small set of basic 3D geometries (a cube, an ellipsoid, a prism, a pyramid and a sphere) with distinctive and expected divergence of geometrical properties. In what follows, the concept of "standard geometries" initially described by Carl et al. (2023) as a geometrical systematization to collect and catalogue subsurface geometries of the potential host rocks in the German site selection for a nuclear waste repository (halite rock, claystone and crystalline rocks) is reviewed, adapted and extended. Please note that the classification is purely geometric, even though the terminology of subdividing categories can also be found in topological considerations (see for instance Thiele et al., 2016a, b). For more details on the classification, please refer to Carl et al. (2023). For real word examples of the standardized geometries, we refer to the publications mentioned in the respective parts of the following paragraph.

Claystones and shales are clastic sedimentary rocks which are commonly deposited conformably onto the underlying strata (Selley, 2000; see also Fig. 1, upper section). The appearance of these conformable layers can vary considerably: tilting and

folding of a flat-lying structure can result in a range of geometries varying from a flat layered appearance that remain generally conformable (see Fig. 2, 4th and 9th row for potential visual representation). By contrast, faulting, erosion and folding can produce unconformable geometries (see Fig. 2, 3rd and 4th row). Lateral stratigraphic pinchout is conformable proximally but results in an unconformity at its tip (see Fig. 2, 4th row). Salt rock (i.e., halite) is initially deposited conformably as an evaporitic sediment. Beyond the undeformed, concordant, flat-layer geometry, halite structures are mainly categorized according to two principles: The most common classification is based on the question whether a structure remained concordant in respect to its overlying rocks or intruded into its overburden (Hudec & Jackson, 2007; see Fig. 1, middle section). Following this systematization, salt anticlines, pillows and rollers are categorized as concordant (see Fig. 2, 1st, 5th and 6th row), while salt stocks, sheets and walls are intrusive bodies (see Fig. 2, 7th to 11th row). In addition, a supplementary subdivision based on the length-to-width ratio of salt bodies is discussed by some authors (e.g., Hudec et al., 2011): Structures showing a length-to-width ratio higher than 2 in map view (thus being considerably anisotropic) are being defined as anticlines or walls, respectively. In contrast, rather isotropic geometries with a length-to-width ratio smaller than 2 are the pillows, stocks and, at least in their early evolutionary stages, sheets. An additional aspect to consider when classifying salt structures is whether the halite is allochthonous or autochthonous. Sheets are the only structural type categorizable as the former: If the bulb of a stock or wall is subhorizontally oriented or moderately dipping above the autochthonous salt source layer, this rock body can be defined as a salt sheet (Hudec and Jackson, 2006). Crystalline rocks considered in the context of the German site selection are plutonic rocks as well as high-grade metamorphic rocks (migmatites and gneisses). As the high-grade metamorphic rocks originate from a wide array of protoliths, resulting in diverse geometries, the establishment of a single, coherent classification for both groups is difficult. For instance, orthogneisses and some migmatites originate from plutonic protoliths such as granitoids and exhibit structural characteristics similar to their igneous predecessors. By contrast, paragneisses and the remaining migmatites derive from various sedimentary sources. Their current shape depends not only on the geometry of the original rock body but also on the specific deformation history experienced during metamorphism. Overall, most high-grade metamorphic rock bodies in the German subsurface are laterally bounded by either plutonic intrusions or fault zones and their top is either bound by unconformities or represents the present-day topography in most cases. Consequently, for the purposes of our geometric approximation, we treat them as discordant rock volumes of varying shape and size (see Fig. 1, lower section and Fig. 2, 2nd, 6th and 8th to 9th row). For plutonic rocks, our classification combines the shape of the bodies with their relationship to the overlying strata (concordance or discordance) (Philpotts & Ague, 2009; see Fig. 1, lower section). Among discordant bodies with varying shape, two size-based categories are distinguished (Fig. 2, 2nd, 6th and 8th to 9th row): batholiths (exceeding $100 \, \text{km}^2$ in areal extent) and stocks (smaller than $100 \, \text{km}^2$). Additionally, cylindrical discordant bodies, mainly representing feeder pipes for ascending magma, are recognized (Fig 2, 10th row). Moreover, two kinds of tabular geometries can be distinguished: discordant dikes and predominantly concordant sills (Fig. 2, 3rd and 8th row). Beyond these, three concordant geometries are noted: laccoliths (characterized by a roughly flat base and a convex roof), lopoliths (defined by a roughly flat top and a shallow convex base), and phacoliths (lens-shaped bodies lacking any flat boundaries; Fig. 2, 5th row).

Building on these classifications, a collection of geometrical end members (standard geometries) that approximate the shape variations of the rock types was set up by Carl et al. (2023). The geometries are intended to act as open source benchmark models for structural geomodeling, as realistic geological models depend on a clear definition of the rock type and the 3D

geometries of evaluated rocks. In its initial form, each of the geometrical end members per potential host rock type was represented by a single version of a 3D body. However, as a large share of these initial end members can be represented by a multitude of possible regular geometrical representations, we designed alternative realizations after reviewing literature: Subsurface salt structures have been created after Hudec & Jackson (2007), Hudec et al. (2011) and Jackson & Talbot (1991), claystone geometries have been inspired by Selley (2000) and Nichols (2009), and crystalline rock geometries are

based on Markl (2015) and Winter (2013). Additional inspiration was drawn from studying open source 3D models of real subsurface structures (Dutch subsurface models from TNO, available at https://www.dinoloket.nl/en/subsurface-models/map, and Australian subsurface models from Geoscience Australia, available at https://portal.ga.gov.au/3d). The standard geometries were created in blender (https://www.blender.org/) and are visible in Fig. 2. Some standard geometries are non-unique for rock types but can be used in different environments, e.g. stocks/batholiths for salt and crystalline intru-

sions. This is indicated in the model titles, as in these cases, the names of different structures are separated by an underscore. Blanks in model names are replaced with a period, and in brackets, additional geometrical information are given in some cases, such as the lateral character of the top of a structure (e.g. hourglass-shape) or the roundness of the top surface (rounded or flattened).

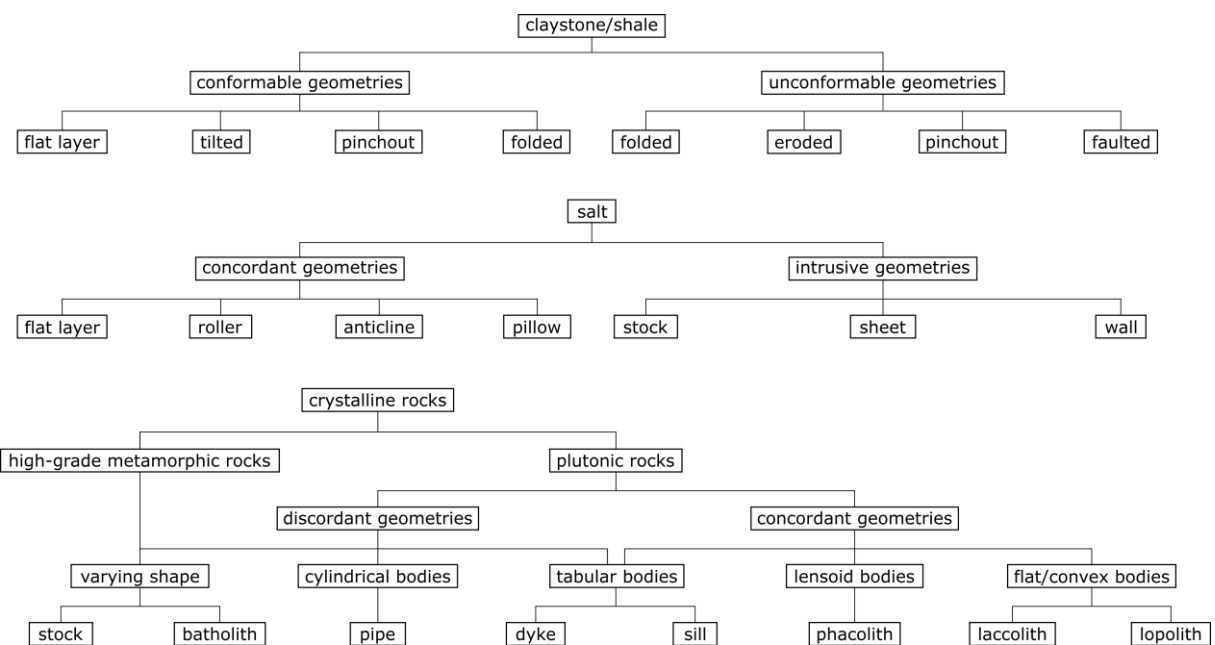

**Fig. 1: Geometrical systematization of the rock types considered for the establishment of the catalogue of benchmark models ("standard geometries"). Adapted after Carl et al. (2023)**

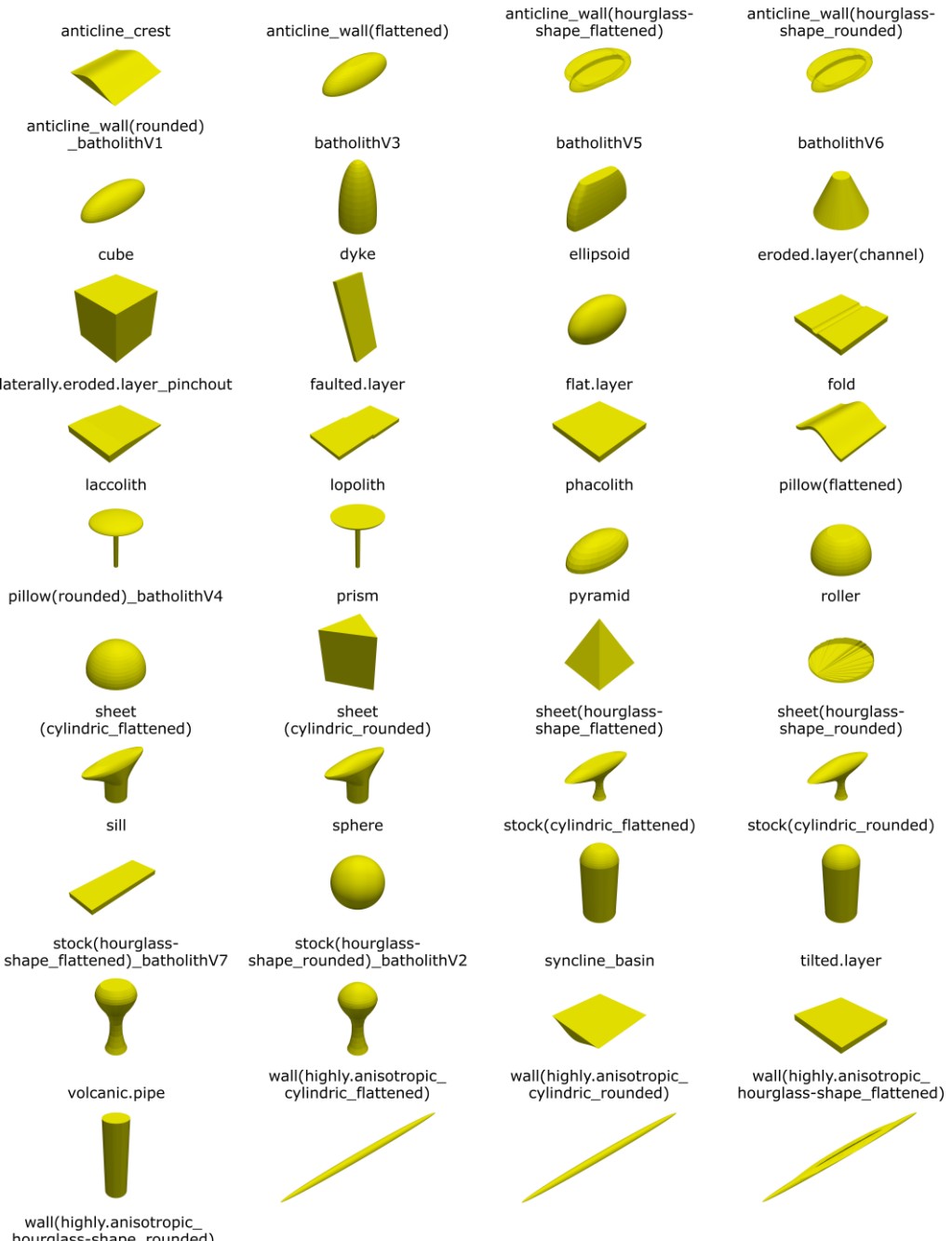

**Fig. 2: Overview of regular synthetic models used in this study. The structures (apart from the cube, ellipsoid, prism, pyramid and sphere) are meant to represent geometrical end members of different rock types ("standard geometries"). For information on the naming convention, please refer to the end of Sect. 1.1. The size of the models was chosen arbitrarily**

**1.2 Geological description of the example model "Altenbruch-Beverstedt"**

The methodology presented herein is illustrated exemplarily on the mesh of the intrusive salt structure Altenbruch-Beverstedt (Lower Saxony, Germany; see Fig. 3). Tectonically, it is located within the roughly N-S striking Glückstadt Graben, developing since the Triassic (Scheck-Wenderoth et al., 2008). The considerably anisotropic salt wall is the result of a complex evolutionary history especially throughout the Mesozoic, as variations in the tectonic regime repeatedly led to shifts from subsidence to uplift in the sub-basins and grabens of the North German Basin (Maystrenko et al., 2008; Scheck-160    Wenderoth et al., 2008; Stollhofen et al., 2008). Within the Glückstadt Graben, the largest salt walls of the German subsurface can be found (Scheck-Wenderoth et al., 2008). The structure Altenbruch-Beverstedt represents a fitting example model for the methodology presented herein, due to its anisotropic, yet complex shape. The anisotropy visualizes well the cross sections created in the first part of the segmentation approach, while the sinusoidal shape illustrates well the segmented nature of the second set of sections (see Sect. 2.1).

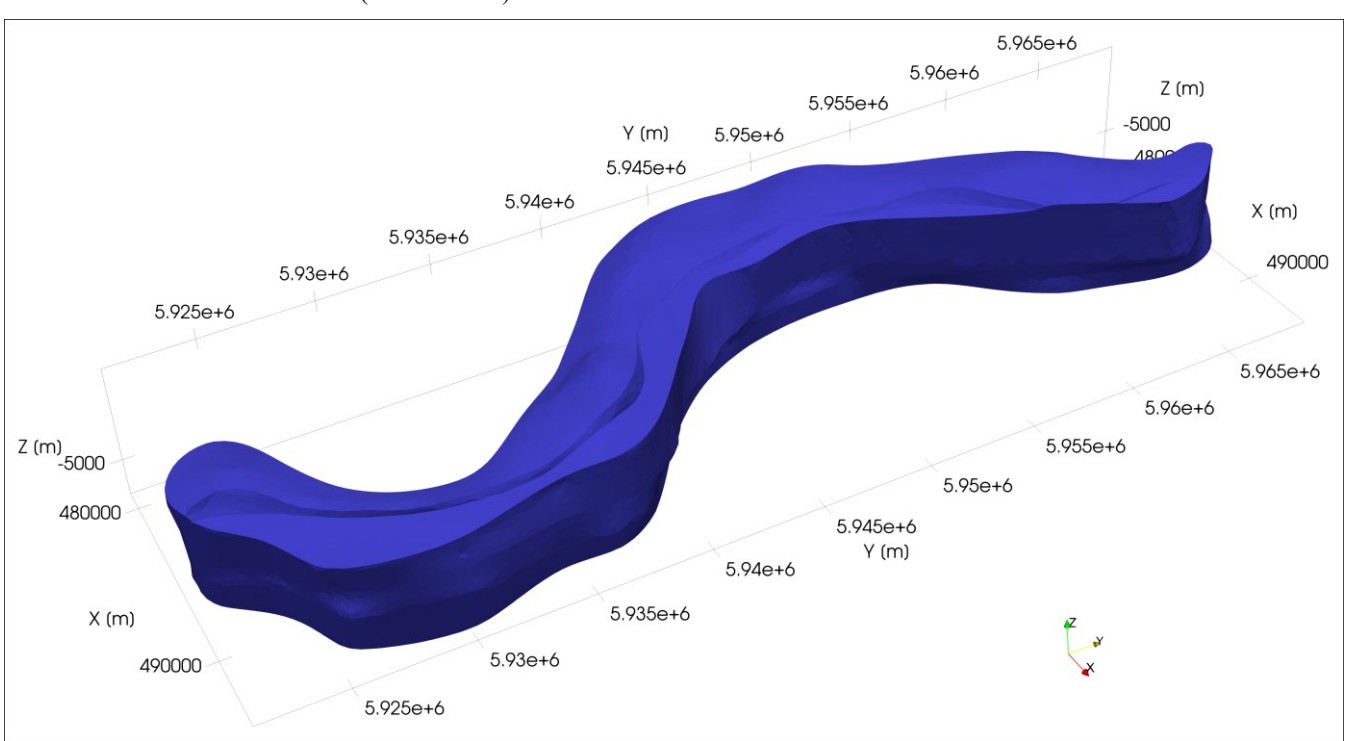


**Fig. 3: 3D model of Altenbruch-Beverstedt, taken from BGR et al. (2022). Coordinate system: EPSG:4647**

## 1.3 Content, motivation and distinction of this study

The proposed methodology allows for the quantitative description, comparison and systematization of explicitly modelled structures and lower dimensional input data using a set of geometrical parameters. The horizontal and vertical dimensions as well as gradients and curvatures of 3D geometries are measured on vertical cross sections oriented perpendicular to the two horizontal main axes of the structures. For 2D inputs (such as 2D geophysical cross sections), alignment of sections is omitted. The resulting datasets are analyzed statistically, providing insight into the main geometrical characteristics of the input

structures: the data analysis yields information about the anisotropy of structures, the potential existence of overhangs, the sphericity and the character of the lateral walls as well as top surface of evaluated structures. Furthermore, K-means clustering is used to systematize the datasets based on the measured parameters. Given 3D input, the setup of cross sections perpendicular to the main axes ensures, that the input structures are covered thoroughly with regular-spaced measurements that follow the 3D contours of the respective geometry. The method is applied to the standard geometries serving as benchmark

geometries for structural modeling of geo-bodies. Applying the method to these models serves two purposes: While the functionality of the approach is validated, we also assess quantitatively, whether the developed benchmark models are geometrically dissimilar or whether some of them can be consolidated for their purpose. Our quantification method represents a rather simplistic approximation approach for the quantitative comparison of 3D structures and lower dimensional datasets that can reproduce the main geometrical characteristics of input datasets fast but also enhances the interpretability of results, making

them accessible to a broader audience. Our method cannot be used to quantitatively compare implicit representations of structures directly from a scalar field, though.

As recognized in Sect. 1.1, similar approaches are rare in general and particularly in geosciences as quantitative approaches commonly aim at uncertainty assessment. In computer sciences, however, Celenk (1995) describes a method involving the alignment of equally-spaced cross sections in two objects via the computation of their respective horizontal main axes and

subsequent section comparison. However, this method is more approximating compared to the proposed approach, as sections are not segmented to align with the contours of the structures.

The proposed method is intended to be utilized in a geomodeling workflow at different stages. 1) Given a sparse dataset including for instance borehole data and 2D seismics of limited quantity, the method can be initially used for a first structural approximation of a targeted geo-body. In the specific example of the German site selection, where most structurally complex

bodies have already been excluded from the considerations (BGE, 2020), this approximation can be achieved using the set of standard geometries that is established in this study. 2) Structural conceptualization and model approximation can also facilitate hyper parameter selection for subsequent interpolation (Wellmann and Caumon, 2018). 3) After the creation of a set of stochastic model realizations, our quantification method and the benchmark models can be applied again in combination with the input data to limit the realizations to the geometrically reasonable ones. However, it has to be noted that this step

would be rather time-consuming for large amounts of realizations. Here, the framework for automatic consistency checking of 3D geological models recently introduced by Parquer et al. (2025) represents a more sophisticated approach. Still, the

proposed framework could reveal model realizations not respecting the conceptual model, which could prompt questions about the assumed geological situation and/or subjective bias, as studied for instance by Bond et al. (2007, 2015). 4) Lastly, the proposed framework can be used for the direction-dependent quantification of modeled structures to assess their potential capacity for material storage (BGE, 2023).

The paper is structured as follows: Sect. 2 outlines the methodology employed in this study, detailing the developed segmentation and measurement algorithm. Sect. 3 presents the results of applying the methodology to the benchmark models and a single subsurface dataset, while Sect. 4 discusses the implications of these findings in the context of existing research. Finally, Sect. 5 concludes with future research directions.

## 2 Methods

### 2.1 Segmentation and measurement algorithm

For our approach, we aimed at a high grade of automation and easy integration in a model analysis process. The method requires the dataset to either be a mesh with extractable vertices or a data frame of vertex coordinates themselves (the system currently only supports .vtk file formats). In what follows, the functionality of the method is explained given a 3D input mesh, but skipping the cross-section generation, the algorithm is also usable for existing cross-sectional data (e.g. geophysical data).

To retrieve characteristic statistics, a geometrical segmentation algorithm (see e.g. Shamir, 2008) has been established, which first discretizes the 3D model into 22 equidistant cross sections with the normal direction parallel to the longer horizontal axis of the mesh´s bounding box. As measurements are conducted perpendicular to the two horizontal main axes of the structures, two sets of cross sections need to be determined separately. Orientation of sections normal to the longitudinal axis of the structure (first direction) have been determined by a minimization of the cross-sectional area, as sections are sequentially rotated (Stephenson, 2018; Fig. 4, Part 1). The cross sections normal to the first set are set up by discretizing the established sections vertically, then first connecting raster lines of consecutive sections and lastly the resulting segments (Fig. 4, Part 3). After their respective setup, the cross sections of both directions are corrected automatically and/or manually for artifacts (Fig. 4, Part 2). Extensional measurements are conducted on each cross section at 5 equidistant transects (Fig. 4, Part 4). Since the very first and last cross section of both directions are excluded from the measurements as they would (undesirably) slice irregular polygons several times, 20 intervals are considered for every input structure. This results in 100 measurements being conducted respectively for each of the two horizontal parameters as well as 200 values for the vertical extent. Please note, that the assumption that a cross section of the first set is perpendicular to the longitudinal horizontal main axis only applies to the center point of the given section. The same limitation applies to a given cross sectional segment (trapezoidal segment) of an orthogonal section and the secondary horizontal main axis.

In addition to the extensional measurements, gradient and curvature calculations are carried out (see Fig. 4, Part 4). Both parameters are determined on all cross sections between consecutive vertices of a cross section. The curvature in 2D is de-

fined as the reciprocal of the circumradius of a triangle. Therefore, it is calculated between three consecutive vertices in either the xz- or yz-plane, by first determining the side lengths (a, b and c) of the triangle between the points, then the semi-perimeter of the triangle and the area through Heron´s formula, before calculating the curvature as the reciprocal of the circumradius of the triangle through:

$$curvature = \frac{4 \times area}{a \times b \times c} \tag{1}$$

The selected method measuring the lateral extents of input meshes normal to their horizontal main axes (see Fig. A1a in appendix) is advantageous over approaches analyzing an input body using parallel sections as applied in various medical imaging techniques like for instance MRI (see e.g. Meyer-Baese & Schmid, 2014). Such an approach would have resulted in a dissimilar amount of output measurements for the two horizontal extents for many input structures as well as for different structures overall, both in case of a uniform regular grid for all datasets (Fig. A1b&e) as well as an individual regular grid per dataset (Fig. A1c&f). Only the usage of an anisotropic grid, depending on the bounds of the input mesh, would have resulted in an equal amount of measurements per horizontal direction (Fig. A1d&g). However, using a supplementary grid would have generally resulted in the problem, that irregular structures would have often been cut several times along a horizontal measuring line. This would have created subordinate polygons that are completely disconnected from each other (see red lines in Fig A1b).

In contrast, covering every input structure with a constant number of measurements as also applied similarly by Celenk (1995) comes with an advantage and a disadvantage: while it ensures that the quantification of input datasets with our method is scale-independent as datasets of different structures have the same amount of data, the geometrical spatial variability of larger bodies might not be captured equally well as the shape of smaller ones. The potential impact of this matter is currently being analyzed in a follow-up study that applies the methodology to a database of over 300 structural models of subsurface structures from various geological settings. The question whether structures shall be represented by equal or dissimilar data quantities also concerns the gradient and curvature data: Orthogonal sections are created from a set of 19 trapezoidal segments (i.e. 40 vertices), while cross sections in the first direction are based on a varying, most often higher number of vertices. As gradients and curvatures are being calculated between neighboring vertices, the potentially larger edges between vertices in the orthogonal sections lead to a less-rounded appearance of the cross sections, directly affecting the values of both parameters.

```
1: Part 1:
2: Input: initial cross sections
3: Output: cross sections rotated perpendicular to longer horizontal main axis by minimization of cross section area
4: for each initial section do
5:      for each of 38 rotation steps (0° to 180° in 5° increments) do
6:          Compute rotation angle (theta)
7:          Apply rotation matrix to original normal ([1,0,0] or [0,1,0]) to get rotated normal
8:      end for
9:      for each rotated normal do
10:         Slice mesh using rotated normal and center point of section
11:         Retrieve vertices from rotated slice
12:         Project cross section into YZ plane to calculate area
13:         Compute centroid of this projected section
14:         Sort points by angle relative to centroid
15:         Compute area of polygon using sorted points and shoelace formula
16:     end for
17: end for
18:
19: Part 2:
20: Input: rotated cross sections
21: Output: rotated cross sections after artifact correction
22: Step 1:
23: for each rotated cross section do
24:     set vertex with lowest z-value as vertex index=0
25:     perform normalized nearest neighbour algorithm
26: end for
27:
28: Step 2:
29: for each section after normalized nearest neighbour algorithm do
30:     apply correction criterion
31:     if correction criterion = True then
32:         initialize manual vertex order correction in plotly.dash
33:         for each section with artifacts do
34:             correct vertex order by clicking on previous (correct) vertex then incorrect vertex
35:         end for
36:     end if
37: end for
38:
39: Part 3:
40: Input: corrected cross sections of 1st direction
41: Output: corrected cross sections of orthogonal direction
42: for each section of 1st direction do
43:     raster section vertically into 22 vertical lines
44:     for each vertical line do
45:         retrieve X, Y and Zmin+Zmax - coordinates
46:     end for
47: end for
48:
49: for 2 consecutive sections of 1st direction do
50:     for all vertical lines in both sections do
51:         extract X, Y and Zmin+Zmax (=2 points per section)
52:         combine 4 points into trapezoidal segment
53:     end for
54: end for
55:
56: for every index of vertical lines do
57:     combine trapezoidal segments to assemble uncorrected orthogonal section
58: end for
59:
60: for uncorrected cross sections of orthogonal direction
61:     repeat Part 2
62:
63: Part 4:
64: Input: all cross sections
65: Output: horizontal and vertical dimensional measurements
66: for each cross section of 1st direction except index 0 & 21 do
67:     rotate & project section onto YZ plane
68:     create 5 horizontal and vertical measurement transects
69:     measure horizontal and vertical dimensions between intersections of transect and polygon
70: end for
71:
72: for each cross section of orthogonal direction except index 0 & 21 do
73:     rescale sections (corresponds to rotation & projection onto YZ plane)
74:     create 5 horizontal and vertical measurement transects
75:     measure horizontal and vertical dimensions between intersections of transect and polygon
76: end for
77:
78: for all cross sections do
79:     compute gradients
80:     compute curvatures
81: end for
```

260

**Fig. 4: Pseudocode of the algorithm that creates the cross sections of both directions and measures the dimensional extents, gradients and curvatures on these sections. For further information, see chapter 2**

## 2.2 Data analysis

The geometrical measurements were combined into a database, analyzed by the first three statistical moments, standard deviation and median and visualized as histograms and cumulative distribution functions (CDF´s). Comparative analyses of data distributions and a cluster analysis were carried out on the measured data, to demonstrate that the tested 3D bodies can be quantitatively compared based on the statistical distributions of geometrical properties and to assess their dis-/similarity. Semi-quantitative comparison of histograms was done for the statistical data, analyzing the vertical extension measures, combined horizontal extension measurements, the gradients and curvatures. For gradient data, the frequency of infinite values was counted separately, since they represent vertical segments between two consecutive vertices. As those values cannot be plotted together with the remaining data as a separate bin, their frequency was visualized as a horizontal line. For gradients and curvatures, overflow bins were established: for the gradient data at the $5^{th}$ and $95^{th}$ percentile and for the curvatures only at the $95^{th}$ percentile. This aimed at facilitating the interpretability of the histograms, since for most datasets, a small percentage of values ($<5\%$) was considerably larger than the rest, thereby spreading the measurements to a large number of additional histogram bins. The Kullback–Leibler divergence (Kullback and Leibler, 1951) was calculated on normalized data between the individual distributions of the geometrical parameters of the input models, for quantification of the similarity between the structures. Cluster analysis followed data normalization to a range of -1 to 1 and principal component analysis (PCA; see Jolliffe, 2002). As variables ("features") for PCA, 20 percentiles of the probability density functions (PDF´s) of the combined horizontal data, vertical data, gradients and curvatures were chosen. As the first two principle components only explained 40% of the variance, a matrix plot for the principal components 1 to 12 was assessed initially, to cover 90% of the variance. A feature angle matrix was then used to reduce the number of principal components in the cluster matrix plot. The number of clusters used in the K-means clustering algorithm was determined using an elbow plot and the silhouette score.

## 3 Results

Results of the segmentation and measurement algorithm as well as the data analysis are demonstrated using a sphere and the intrusive structure "Altenbruch-Beverstedt" (model taken from BGR et al.,2022). Subsequently, the results of the cluster analysis are presented.

### 3.1 Segmentation and measurement algorithm

The initial subdivision of the input mesh (Fig. 5a & b) is followed by the stepwise rotation of the initial cross sections. The respective rotation step showing the minimal cross-sectional area is optimally oriented normal to the longitudinal main axis of the structure (first direction). Optimal orientation of all sections of the first direction of the sphere corresponds to 0° rotation, unlike when running the algorithm on an irregular mesh like Altenbruch-Beverstedt. This is the case due to the regularity and symmetry of all test models of this study. After subsequent artifact correction (Fig. 5c & d), the second set of cross sections is assembled from trapezoidal segments (for illustration, a subset of sections is shown in Fig. 5e & f).

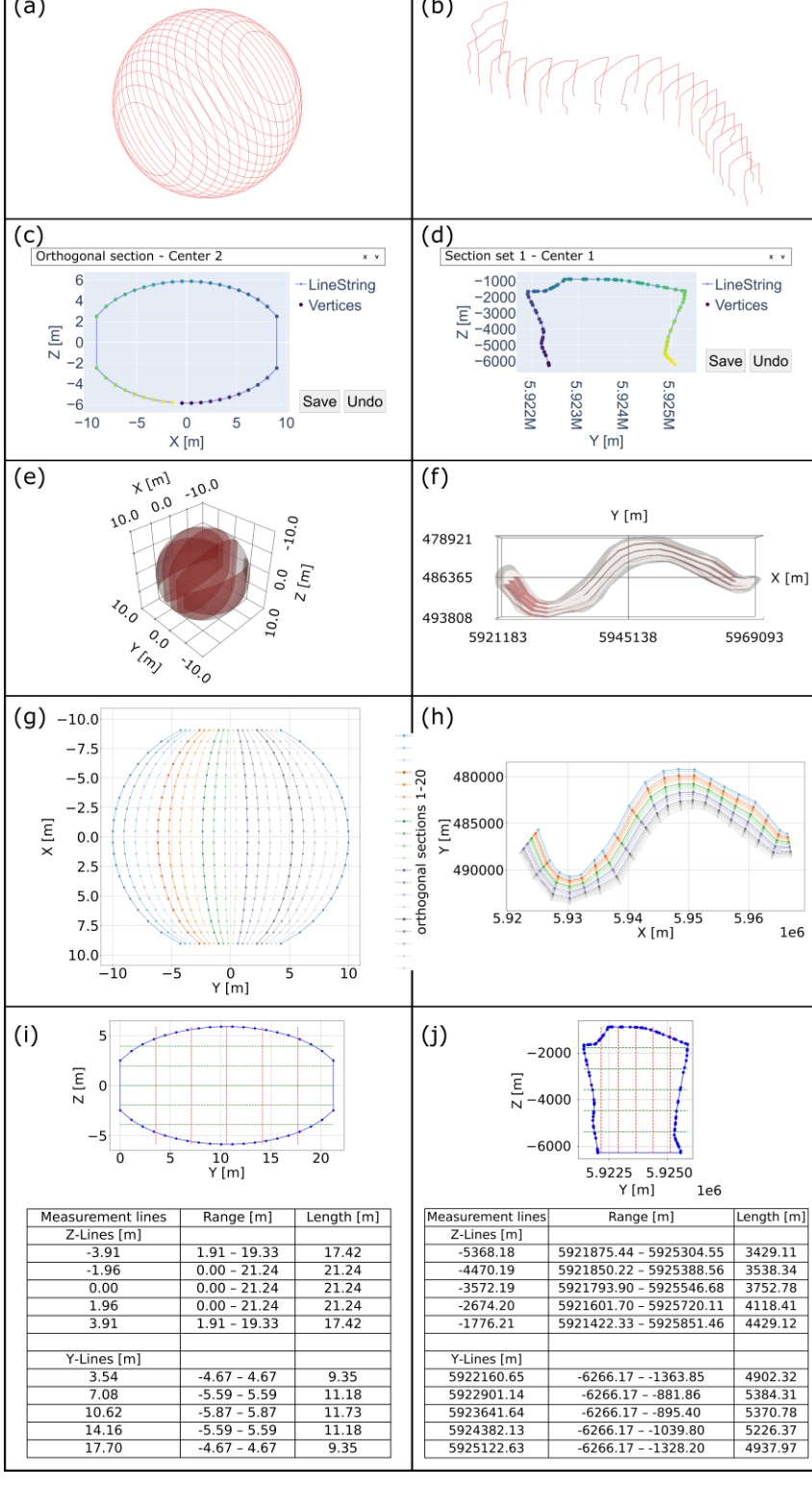

**Fig. 5: Visual representation of the segmentation and measurement algorithm for a sphere model (left column) and the German intrusive salt structure "Altenbruch - Beverstedt" (right).**
**a & b)** initial segmentation of the input meshes.
**c & d)** Plotly.dash app for vertex-order correction.
**e & f)** Subset of orthogonal cross sections.
**g & h)** Coverage of input structure with cross sections (top view)
**i & j)** Example of extensional measurement results

Following potential artifact corrections of the orthogonal sections, both sets of cross sections are finalized (Fig. 5g & h) and extensional measurements as well as gradient and curvature calculations are carried out (Fig. 5i & j). The computational power required by the algorithm is low (runtime without varying artifact corrections: ca 60 s, using as the CPU an AMD Ryzen 7 PRO 5850U at max. 50% capacity at 3.5 GHz speed and the integrated GPU at 0.6 GB usage).

## 3.2 Data analysis

The results of the first analysis step, the first three statistical moments, standard deviation and median per parameter and the data visualized as histograms and CDF´s, are seen in Table 1 and Fig. 6, respectively. The size of the sphere was chosen arbitrarily, as the subsequent Kullback-Leibler divergence and cluster analysis are based on normalized data. Both the statistical moments for the sphere and the distributions in Fig. 6 (left column) reveal differences for the three parameters, although individual extents should be the same in all three dimensions, if a sphere would be measured equally in all directions. This is due to compromises of the algorithm ensuring its universal applicability. For Altenbruch-Beverstedt, the large variance and standard deviation of the combined horizontal data and the difference between the mean values of both individual horizontal parameters reflect the strong anisotropy of the structure, while the statistics for the vertical data indicate a moderate variation in vertical measurements.

Table 1: a) First three statistical moments, standard deviation and median per parameter for the sphere model (note: the dimensions of the sphere are chosen arbitrarily). b) First three statistical moments, standard deviation and median per parameter for the model of the real subsurface structure (Altenbruch-Beverstedt). Statistics for Altenbruch-Beverstedt reflect the strong anisotropy of the structure

| (a) sphere | mean [m] | variance [m] | std_dev [m] | skew | median [m] |
|---|---|---|---|---|---|
| Horizontal length | 14 | 13 | 4 | -0.4 | 14 |
| Horizontal length orthogonal | 18 | 4 | 2 | 0 | 18 |
| Vertical length | 14 | 13 | 4 | -0.5 | 15 |
| Horizontal data combined | 16 | 13 | 4 | -0.8 | 17 |

| (b) Altenbruch-Beverstedt | mean [m] | variance [m] | std_dev [m] | skew | median [m] |
|---|---|---|---|---|---|
| Horizontal length | 3825 | 635352 | 797 | -0.2 | 3795 |
| Horizontal length orthogonal | 48644 | 8956682 | 2993 | -2.2 | 49845 |
| Vertical length | 4766 | 486158 | 697 | -2.9 | 4856 |
| Horizontal data combined | 26235 | 5.07E+08 | 22516 | 0.02 | 21696 |

Gradient and curvature histograms of the example cases are visible in Fig. 7. For the sphere, the distribution of the gradient histogram is symmetric (Fig. 7a). The curvature histogram (Fig. 7c) shows a prevalence of very small values and subordinate maxima around 0.1, 0.2 and in the overflow bin that contains 394 values (5% of all data) above 0.37. For Altenbruch-

Beverstedt, the gradient distribution is asymmetric and the number of infinite gradients is higher (Fig. 7b). In comparison to the curvature distribution of the sphere, the curvature data (Fig. 7d) is monomodal apart from the overflow bin.

In general, analyzing the data distributions of a structure visually already reproduces distinct geometrical characteristics of an input dataset. The distribution of the combined horizontal data indicates whether a pronounced anisotropy is present for an analyzed structure: if the data is separated into two clearly distinguishable subordinate distributions (see Fig. 8a), the geometry is considerably anisotropic (the farther apart the two maxima, the more anisotropic a body is). Caution is advised for a distribution with two close maxima (Fig. 8b): this could be the consequence of the inflated extent in the orthogonal

direction (see Sect. 4.2). Analyzing the combined horizontal data and the vertical data together reveals whether a structure shows substantial variations in its horizontal extent over its vertical range. Such a shape, in the subsurface more often present as overhangs rather than as upward tapering, is indicated by the simultaneous presence of multimodal distributions for both parameters (Fig. 8c & d). The vertical data distribution also characterizes the top surface of a geometry: if the distribution is monomodal, with a) the maximum being the bin representing the highest measurements, and b) the frequency in lower bins

being substantially smaller, then the presence of a flat top surface is indicated. The existence of a flat top surface can be verified by analyzing the gradient and curvature data: a high frequency of very small measurements for both parameters supports such an analysis (Fig. 8e-g). Gradient data also indicates the steepness of lateral surfaces of a body: as high and infinite gradient data stem from steep to vertical faces of a structure, the presence of steep-dipping lateral surfaces can be recognized (Fig. 8h). Combining the inferences from analyzing top and lateral surfaces therefore provides insight into the

overall sphericity of an input dataset: a more spherical structure is represented by larger quantities of intermediate gradient measurements and of moderate to high curvature data (Fig. 8i & j).

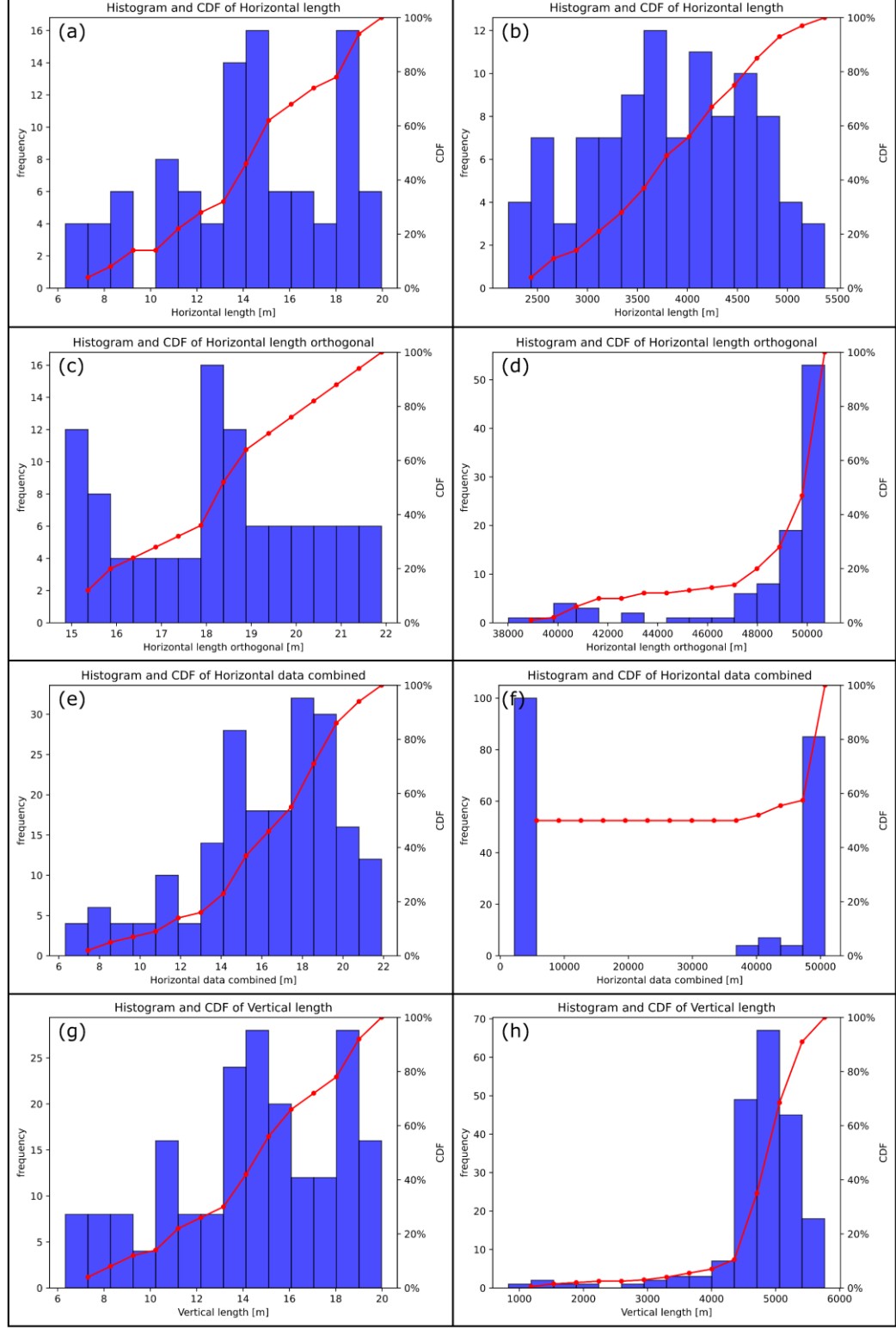

**Fig. 6:** Data distributions and cumulative distribution functions (CDF´s) for the extensional parameters. Left column: sphere, right column: Altenbruch-Beverstedt.
a & b) Horizontal data from the first direction.
c & d) Horizontal data from the orthogonal direction.
e & f) Combined horizontal data.
g & h) Vertical data

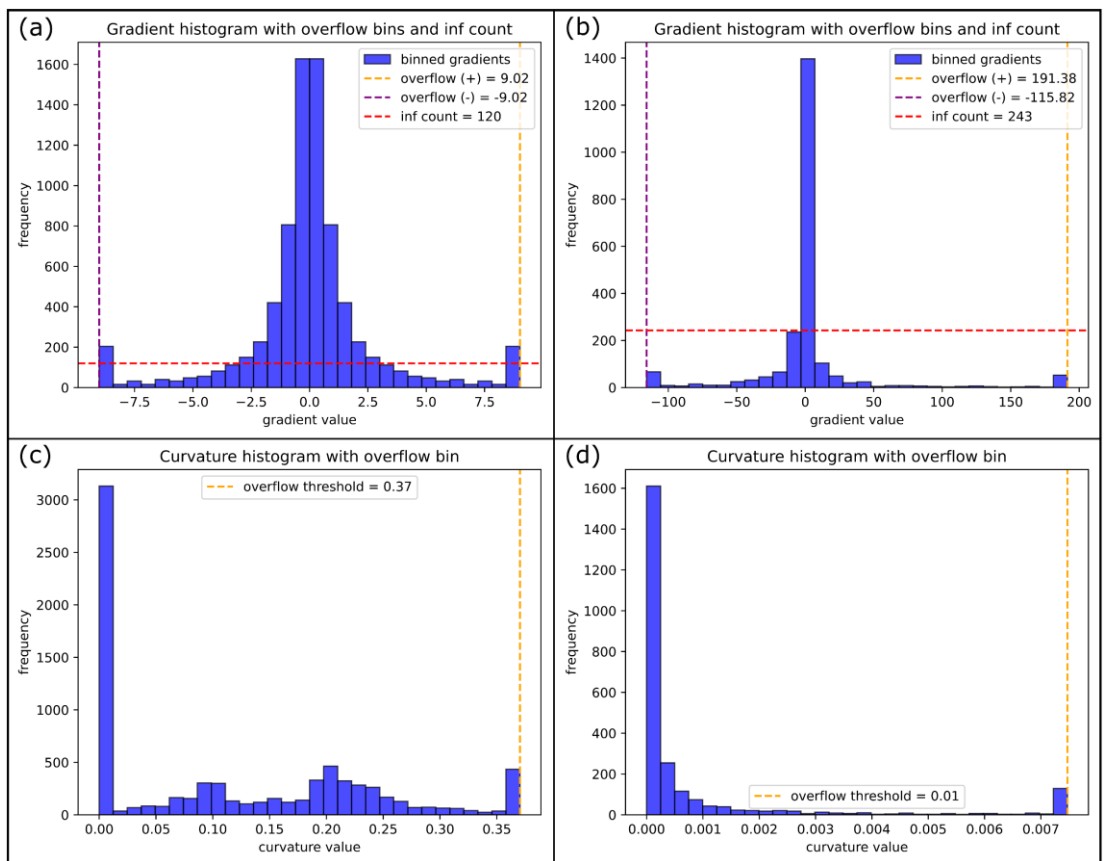

**Fig. 7: Gradient and curvature data for the sphere (a & c) and Altenbruch-Beverstedt (b & d). Amount of data in overflow bins:** Gradient diagram (sphere) 384 values (4.8% of all data), Curvature diagram (sphere) 394 values (5%); Gradient diagram (Altenbruch-Beverstedt) 112 values (4.6%); Curvature diagram 101 values (4.1%)

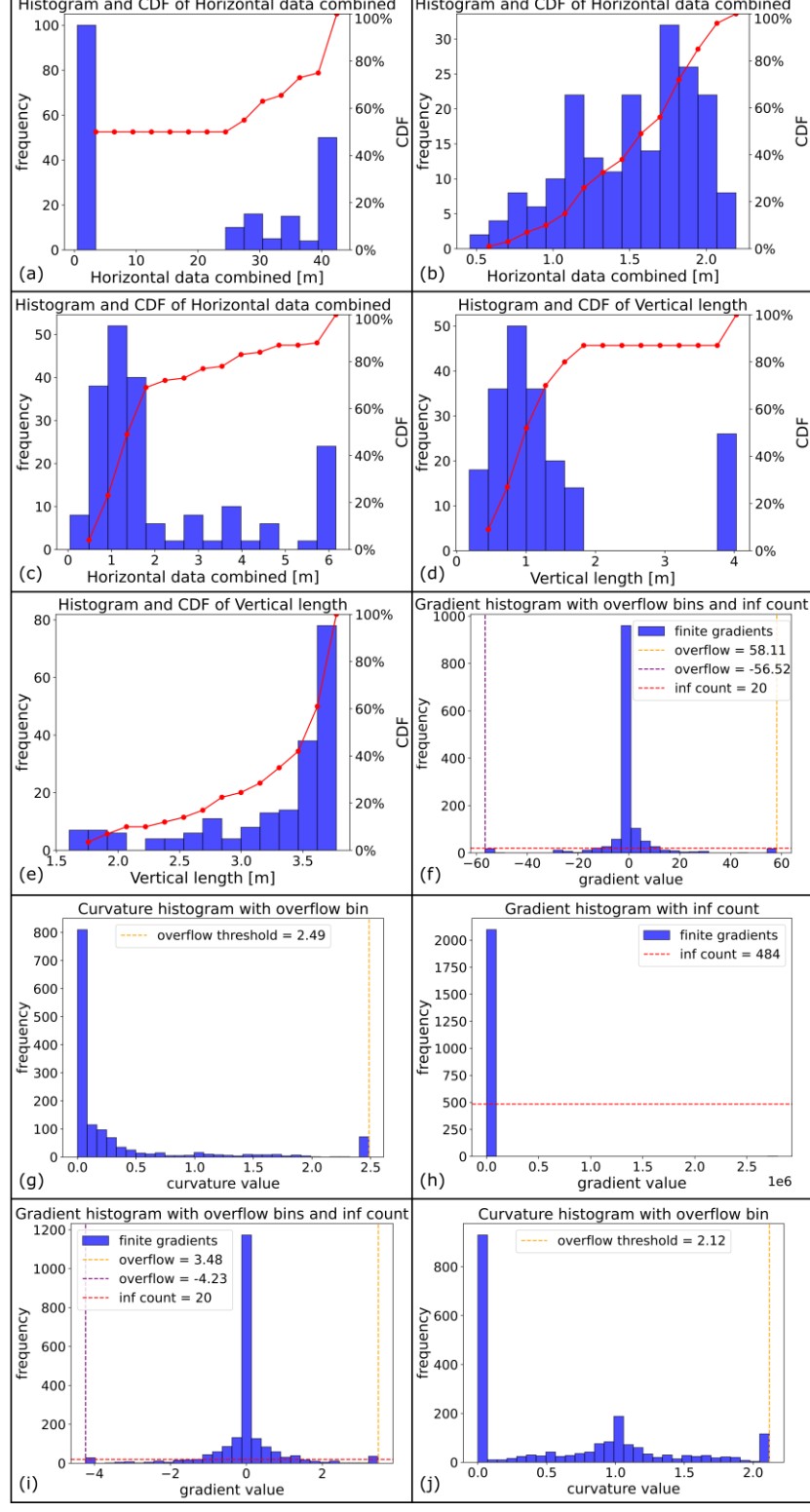

**Fig. 8: Analysis of data distributions to reproduce geometric characteristics of input models. Please compare with model appearances in Fig. 2.**

**a) Combined horizontal data of "wall(highly.anisotropic_hourglass-shape_rounded)", reflecting strong anisotropy.**

**b) Combined horizontal data of "batholithV3" incorrectly indicating slight anisotropy.**

**c) & d) Combined horizontal data and vertical data of "sheet(hourglass-shape_rounded)" indicating presence of overhangs.**

**e)-g) Vertical data, gradients and curvatures of "batholithV5", revealing the presence of a flat top surface**

**h) Gradient data of "volcanic.pipe", reflecting the prominence of vertical lateral walls**

**i) & j) Gradient and curvature data of "pillow(rounded)_batholithV4" showing the spherical character of the input model**

The Kullback-Leibler (KL-)divergence (Kullback and Leibler, 1951) was calculated to quantitatively determine the similarity between the tested geometries. The distributions of the six parameters (the individual horizontal extents along both horizontal main axes, the combined horizontal data, the vertical data, gradients and curvatures) were compared between the models. The similarity of two distributions is larger, the smaller the KL divergence is, with a value of 0 indicating equality of the distributions (obtained for instance when comparing a structure with itself). The result of the calculation of the individual KL divergences for the example cases is visualized in Fig. 9. For the sphere, the most similar models regarding the respective distributions of the six parameters are the "sheet(cylindric_rounded)" for the horizontal data of the first direction, the "prism" for the orthogonal horizontal data, the "batholithV3" for the combined horizontal data, the "anticline_wall(rounded)_batholithV1" for the vertical data, the "phacolith" model for the gradients and the "ellipsoid" for the curvatures (compare Fig. 2 for the model appearances). For Altenbruch-Beverstedt, "batholithV6" is most similar regarding the horizontal data of the first direction, "roller" for the orthogonal horizontal data, "wall(highly.anisotropic_hourglass-shape_flattened)" for the combined horizontal data, "wall(highly.anisotropic_hourglass-shape_rounded)" for the vertical data, "pillow_flattened" for the gradients and "roller" for the curvatures. In addition to KL divergences of individual parameters, an averaged KL divergence was calculated: by taking the mean of the values between two models, the overall dis-/similarity between models was assessed. According to the averaged KL divergence, the sphere is closest to the standard geometry "pillow(rounded)_batholithV4", while Altenbruch-Beverstedt is best approximated by the "wall(highly.anisotropic_hourglass-shape_rounded)". However, informational content of this parameter is limited, as there is no indication regarding which parameters two compared structures are most similar or differ more. Therefore, principal component analysis and K-means clustering have been employed as well, providing this information based on all combined parameters.

In general, values of KL divergence show an error for the gradient distributions: infinite values had to be converted to the highest finite gradient value of a given dataset to enable the computation, inflating the highest bin. Furthermore, the large variance of curvature data for most input models (see for example Fig. 9f & l and Sect. 4.2) decreases the applicability of the KL divergence for that parameter, as most models show very similar normalized distributions. To assess the impact of the large variance on individual KL divergences of curvature data and smallest averaged KL divergences, they were also calculated using a 95$^{th}$ percentile overflow bin (see Table 2). Smallest KL divergences for the curvatures of the two example models are notably higher, especially for the sphere, reflecting the dissimilarity of data distributions when applying the filter (column 1 & 2). The impact on the smallest averaged KL divergence (column 3 & 4) is smaller, yet still considerable.

**Table 2: Comparison of KL divergences with and without the usage of a 95$^{th}$ percentile overflow bin for the curvature distributions.**

| structure | smallest KL divergence for curvature without overflow bin | smallest KL divergence for curvature with overflow bin | smallest averaged KL divergence for all properties without overflow in curvature | smallest averaged KL divergence for all properties with overflow in curvature |
|---|---|---|---|---|
| sphere | 0.0054 ("ellipsoid") | 0.16 ("wall(highly.anisotropic _cylindric_rounded)") | 0.45 ("pillow(rounded)_batholith V4") | 0.56 ("batholithV3") |
| Altenbruch-Beverstedt | 0.036 ("roller") | 0.05 ("batholithV5") | 1.2 ("wall(highly.anisotropic_hourglass-shape_rounded)") | 1.3 ("wall(highly.anisotropic_hourglass-shape_rounded)") |

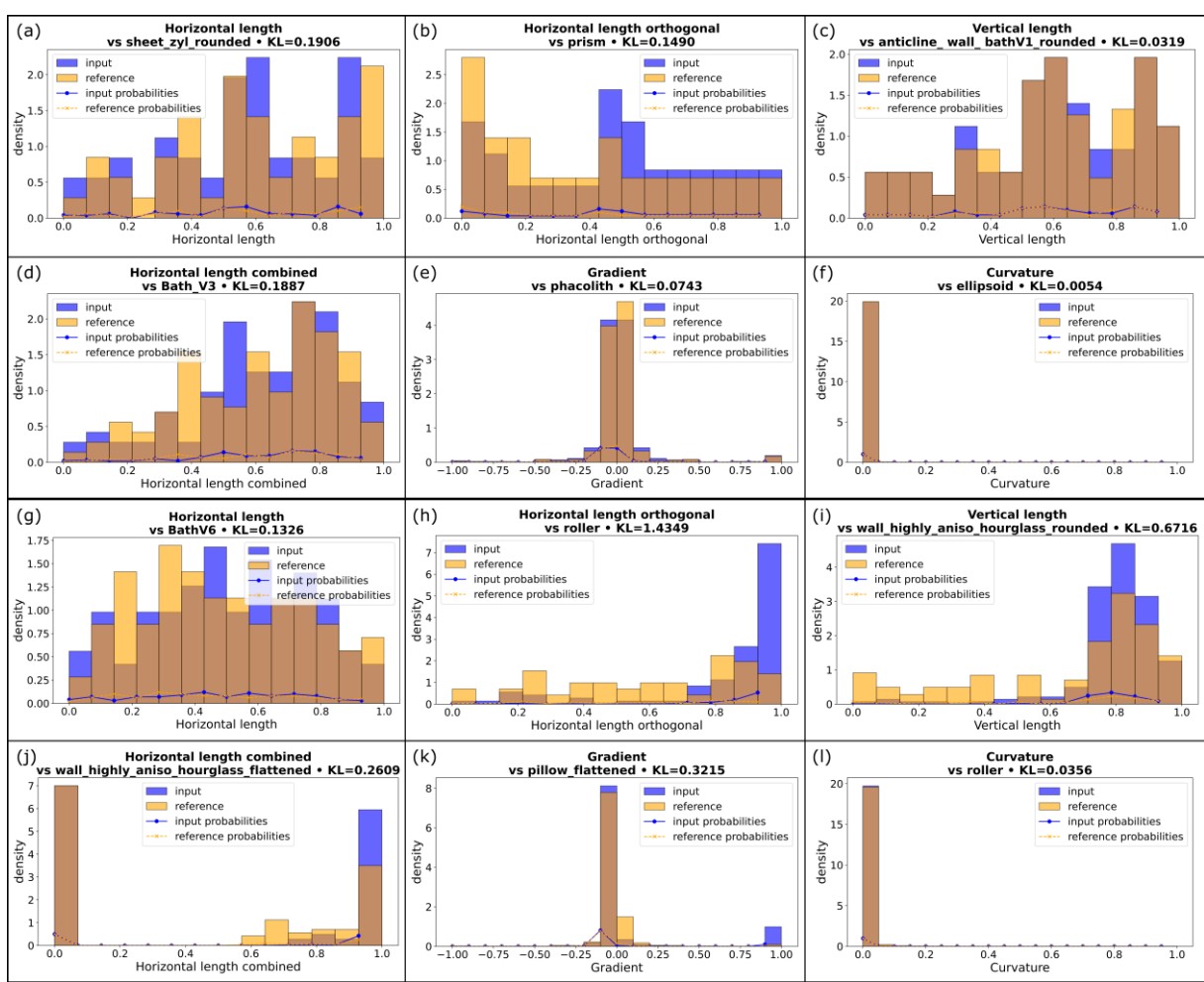

**Fig. 9: Visualized Kullback-Leibler divergences. a)-f): sphere, g)-l): Altenbruch-Beverstedt. Visualized is the most similar data distribution (orange) compared to the respective distribution of the two example models (blue). The calculated Kullback-Leibler divergences are noted in the headlines of individual figures**

## 3.3 Cluster analysis

Cluster analysis on all measured data of the regular geometries resulted in 7 clusters considering the combined analysis of the elbow plot and silhouette score (Fig. 10b). With the first two principle components (PC´s) only explaining 40% of the variance (see Fig. 10a), the number of PC´s necessary to cover more than 90% of the variance was determined to be 12. A feature angle matrix (Fig. A2) was computed to check the dependencies between the percentiles of the PDF´s. As a strong dependency was identified within several groups of features, the principal component matrix plot (Fig. 11) was limited to the

first six PC´s. The feature contribution matrix (Fig. 10c) reveals the contribution ("loadings") of the binned PDF´s to the principal components, with bright yellowish colors indicating a strong positive contribution and dark blue colors a substan-

tial negative contribution. In the contribution matrix, percentiles 0 to 19 represent the PDF of the combined horizontal data, followed by the vertical data (20 to 39), gradients (40 to 59) and curvatures (60 to 79).

The PC1 vs. PC2 cluster plot is visible in Fig. 11a. Positive contributions to PC1 are dominated by the 50 to 55% bins of the vertical data and gradient data, while there is no percentile with a strong negative contribution to PC1. This effectively separates the bluish-green cluster at high positive PC1 scores from the rest of the data. All four models ("flat.layer", "sill", "cube" and "prism"), share a distinct geometrical similarity: When segmenting them with our algorithm, cross sections are always flat at the top and of exactly the same vertical extent throughout the entire structure. This results in a step-wise appearance of the respective PDF´s, with the step being in the middle of the functions. For PC2, the 45 to 50% gradient bin has the highest positive loading, while there are stronger negative loadings for the 0 to 5% as well as 95 to 100% gradient bins. This separates the models of the black cluster at highest positive PC2 scores and mainly the blue cluster at high negative PC2 scores. Therefore, models of the black cluster are characterized by the presence of many low to moderately inclined surfaces in a geometry (depending on the variance in a data distribution) and an overall more rounded appearance (see e.g. the highlighted black example model "pillow(rounded)_batholithV4" in Fig. 11a). Meanwhile, the blue models and other models at high negative PC2 scores are characterized by the abundance of steep-dipping to vertical surfaces. Thus, PC2 is an indicator for the overall steepness of the lateral parts of a structure or, on the other hand, its sphericity.

The PC3 vs. PC4 cluster plot is shown in Fig. 11b. For PC3, large positive contributions are spread among the 0 to 5% and 95 to 100% horizontal bins as well as the 95 to 100% vertical bin and the 0 to 5% and 45 to 50% gradient bins, while the only considerable negative loading is exhibited by the 10 to 15% gradient bin. Datasets at very positive PC3 scores belong to the vermilion and blue clusters. The largest negative PC3 scores are seen for the reddish-purple cluster. Very positive PC3 scores indicate anisotropy, rather flat top surfaces and steep-dipping to vertical lateral walls (see e.g. the highlighted vermilion "dyke"). In contrast, however, datasets at largest negative PC3 scores, cannot be linked to very high data percentages in that 10 to 15% gradient bin; its loading (-0.27) not being the main cause of the observed negative PC3 scores. PC4 shows considerable positive loadings for the 50 to 55% bin of the vertical data and the 0 to 5% and 95 to 100% gradient bins. Meanwhile, large negative loadings are seen for the 0 to 5% and 95 to 100% horizontal bins, the 95 to 100% vertical bin and the 80 to 85% gradient bin. These contributions mainly drive the differentiation of the reddish-purple and vermilion clusters (negative PC4 scores) from the other clusters apart from some sky-blue models. Since the horizontal and vertical bins contributing very negatively are the same horizontal and vertical bins contributing particularly positively to PC3, it can be deduced that the overall position of the vermilion models in the PC3 vs. PC4 diagram is more driven by these horizontal and vertical bins. Meanwhile, the datasets from the reddish-purple and sky-blue models are comparatively influenced more by the 80 to 85% gradient bin also showing a considerable negative loading. Still, most datasets from these clusters at negative PC4 scores can be considered as rather anisotropic geometries with mainly steeper (but not vertical) lateral walls, while models at higher positive PC4 scores exhibit uniform vertical extents and steep-dipping to vertical lateral walls. This explains the position of the isolated blue model at highest positive PC4 scores ("volcanic.pipe"; highlighted in Fig. 11b; see also Fig. 8h for the gradient distribution), completing the separation of the blue cluster from the rest of the data.

The cluster plot of PC5 vs. PC6 can be seen in Fig. 11c. PC5 shows strong positive contributions for the 0 to 5% vertical bin and the 50 to 55% and 95 to 100% gradient bins, while stronger negative loadings are given by the 50 to 55% vertical bin and the 0 to 5% gradient bin. This separates the majority of the sky-blue cluster (highest positive PC5 scores) from the rest of the datasets. As this corresponds to the first appearance of the 0 to 5% vertical bin among considerable contributing bins,

most of the associated models are characterized by widespread low vertical extents and few much larger ones, as seen in overhang configurations (for example, see the model "laccolith" in Fig. 2). PC6 is mainly influenced by the gradient data, where the 10 to 15% bin contributes the most negatively and the 80 to 85% bin contributes positively. Once again (as for PC3), the 10 to 15% gradient bin, however, does not seem to be the main reason for the separation of the reddish-purple cluster at very negative PC6 scores. Similarly, the sky-blue models at higher positive PC6 scores do not exhibit particularly

large high percentages in the respective bin.

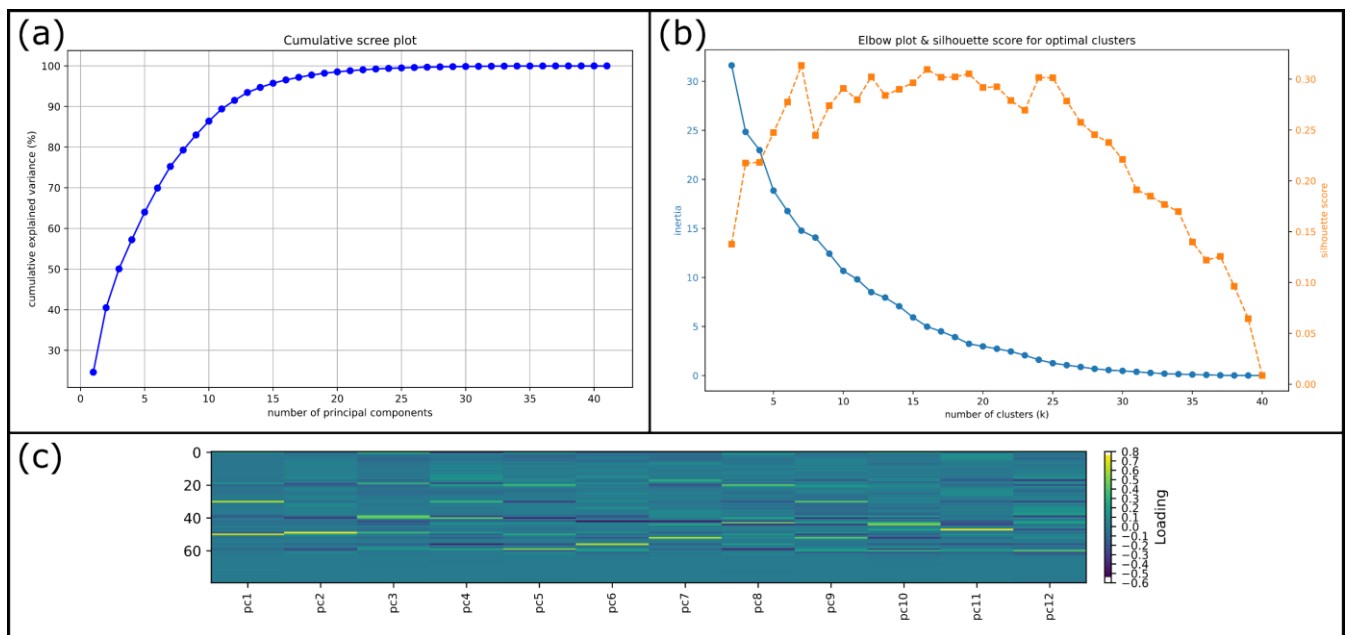

**Fig. 10: Calculated supplementary information for the setup and interpretation of the cluster analysis after principal component analysis. a) Cumulative scree plot, showing the explained variance with increasing number of principal components. b) Elbow plot**
**and silhouette score to determine number of clusters. c) Contribution matrix showing the contribution of the input data (percentiles of the probability density functions of measured parameters) to the principal components**

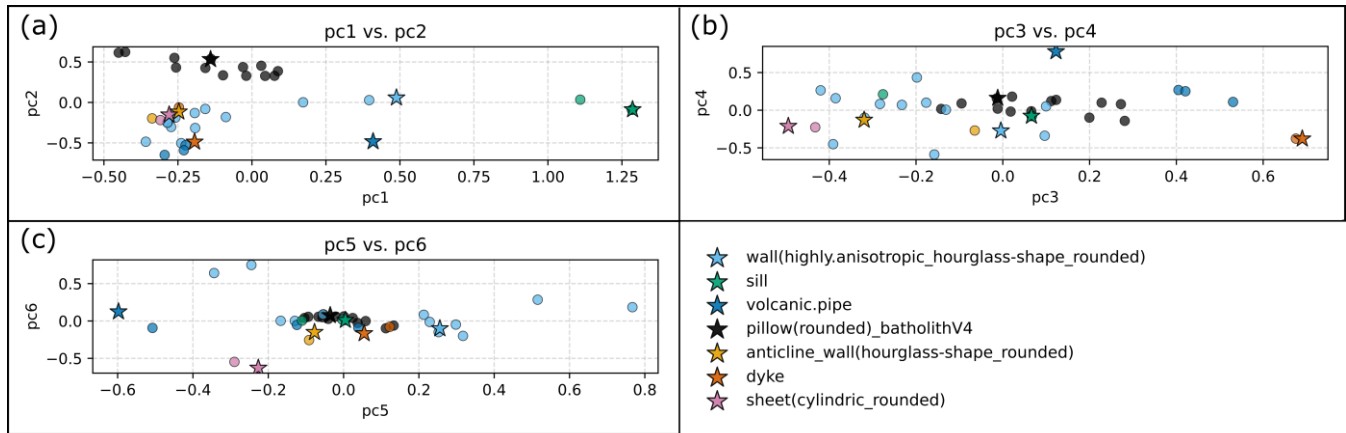

**Fig. 11: Matrix plot of principal components (PC´s) explaining 70% of the variance in the data (see also Fig. 10a). Larger stars mark example models for clusters (see legend)**


The overall cluster results validate that the flat and/or cuboidal geometries (vertical extent ≤ horizontal extent and/or exclusively straight lateral surfaces) mostly differ considerably from the other structures designed to represent intrusive subsurface bodies: the flat/cuboidal geometries are mainly distributed among the bluish-green, blue and vermilion clusters (see Fig. 11). As recognized above, these three clusters can be differentiated from the other models within the first four PC´s. Only the

"laterally.eroded.layer_pinchout)" is located outside, in the sky-blue cluster, although representing a flat geometry. The K-means cluster analysis furthermore indicates that some standard geometries are similar across all parameters. Therefore, it was assessed whether certain standard geometries are redundant to simplify the benchmark selection process. The pairs of flattened and rounded versions show high similarities, leading to the exclusion of the flattened models while retaining the "uneroded" structures. The models "batholithV3" and "pillow(rounded)_batholithV4" cluster closely, differing only in verti-

cal elongation; thus, "batholithV3" is excluded. Although similarities exist between models with varying lateral characteristics, both "cylindric" and "hourglass" shape variations are retained. This also applies to various "sheets" and the "anticline_wall(hourglass-shape_rounded)", which exhibit similar PC scores in some, but not all cluster plots. Lastly, some of the flat and cuboidal bodies in the bluish-green cluster ("flat.layer", "sill" and "cube") are nearly identical in position. The "cube" is excluded from the benchmark collection, while the other two geometries are merged, keeping the shape of the "sill".

Given these exclusions based on structural similarity, the collection is condensed from 36 to 25 standard geometries (see Fig. 12). Decreasing the database by validating the bodies' geometrical dissimilarity facilitates the decision making on the best suitable benchmark for a case study. Despite our reduction efforts, this list is not expected to be exhaustive: we would like to encourage users to suggest additional geometries based on their expertise and/or literature, to ensure that suitable benchmark models are available for as many geomodeling applications as possible.


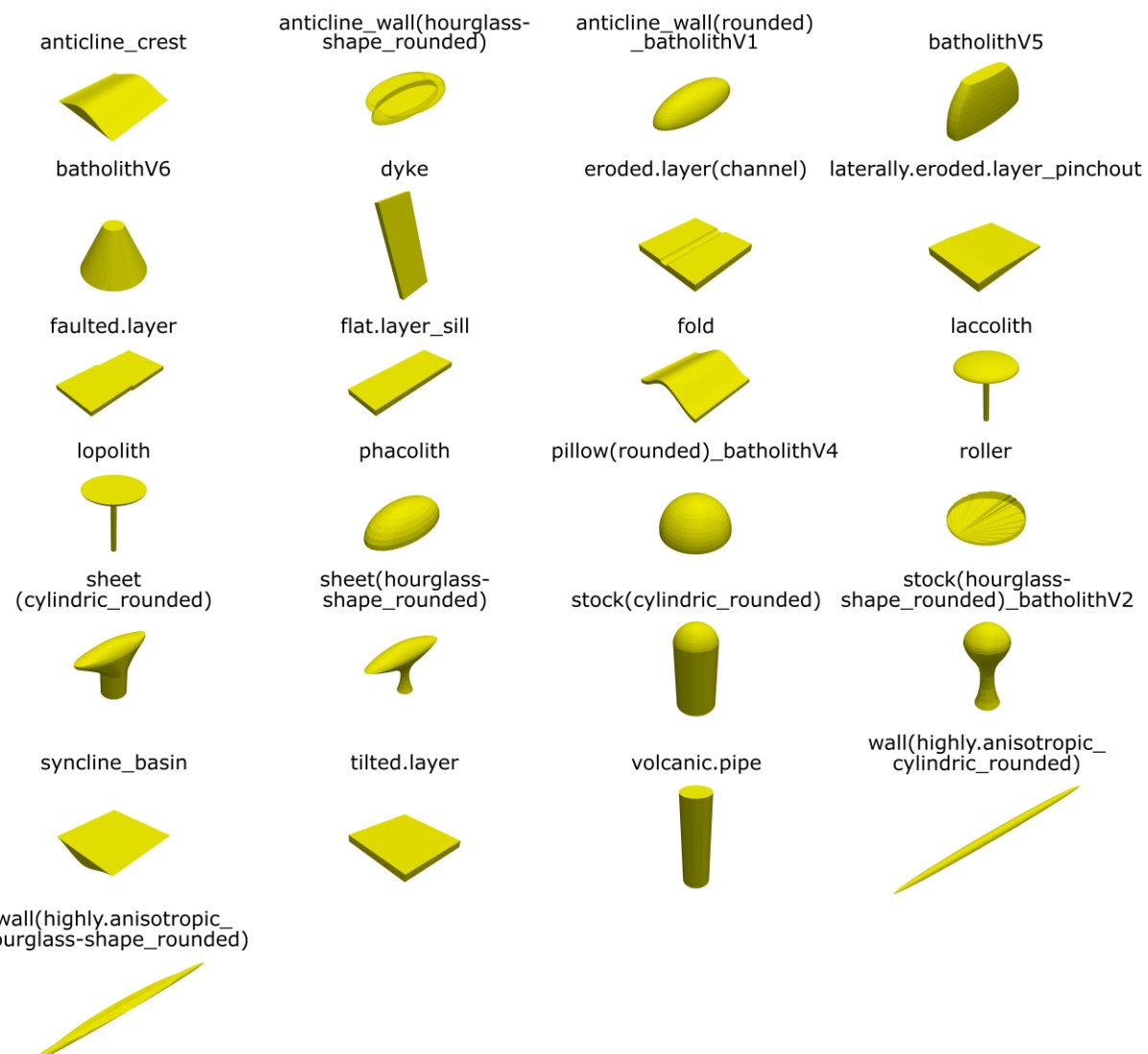

**Fig. 12: Condensed collection of standard geometries after application of the quantification method.**

## 4 Discussion

By applying a set of defined geometrical descriptors to systematically generated 3D benchmark models, this study establish-
es a framework for the quantitative comparison of shape properties. The analysis highlights how key attributes such as ani-
sotropy, surface morphology, and sphericity vary across models, offering a structured perspective on their geometric dis-
/similarities. These outcomes prompt a deeper discussion of how well the proposed descriptors capture meaningful shape
differences and how this quantitative framework advances the analysis of unmodeled data and 3D geological structures.

**4.1 Comparison of existing methods for 3D shape characterization with the proposed workflow**

The workflow of this study differs considerably - both in the scale of the test subjects as well as the purpose of the methods - from the quantitative comparative approaches used in material sciences (Sect. 1.1). The material-scientific studies mainly operate on millimeter- to centimeter-scales and have a higher emphasis on parameters exploring the sphericity/angularity (or similar metrics) of objects, as these characteristics are fundamental in this field, where properties and applicability of composite building materials heavily depend on mechanical interactions between individual particles (Kakani and Kakani, 2004).

Meanwhile, the proposed approach aims at characterizing 3D structures and 2D sections at the meter- to kilometer-scale (although the applicability is scale-independent in theory), through direction-dependent measurements of geometrical parameters, thereby providing datasets suited for quantitative comparison.

Studies presenting approaches that show similarities to ours are Celenk (1995), Schweizer et al. (2017) and Lindsay et al. (2013). Celenk (1995) determines the horizontal main axes of equally-spaced cross sections as well, but does so to align

sections of two different objects. Comparison is then achieved by computing the averaged shape difference of sections between the objects in four directions along the main axes. Key differences of our approach therefore involve the segmented assemblage of cross sections in the orthogonal direction (following the respective segmented horizontal main axis) and the exact measurement of the dimensional extents on the sections. Hence, our method opts for the determination of larger datasets of absolute measurements on a single object, that are compared to other bodies in subsequent steps. Meanwhile,

Celenk (1995) computes the relative measure that is the averaged shape difference, representing a faster, but more approximate approach of object comparison, as the author does not segment the horizontal main axis along the larger extent. Schweizer et al. (2017) do not try to compare the dimensions of individual 3D structures, but use the Jaccard distance and the normalized city-block distance as measures for model dissimilarity instead. The two parameters are being applied as measures for the similarity in position of certain geological units between two model realizations of the same study site. In a

similar fashion, the Hausdorff distance has been used before (see e.g. Suzuki et al., 2008). These dissimilarity distances were not applied in our study, as they could only act as size indicators rather than shape descriptors and would not give any indication on where two structures differ spatially. Meanwhile, our approach provides insight into both shape and size differences of objects, which is crucial for geological modeling. Lindsay et al. (2013) explore geometric uncertainty across multiple realizations of a study site, evaluating parameters like depth, volume, and curvature, which parallel those in our study.

However, their parameters are often tied to stratigraphic units and may not apply directly to individual 3D structures. Both studies utilize PCA to analyze geometric variability and model differences, although executed differently.

Despite these existing methodologies, we opted for a straightforward approach, allowing us to efficiently replicate the main geometric characteristics of input datasets. Our algorithm is computationally efficient, easily interpretable with basic geological knowledge, and accessible to a non-specialist audience.

## 4.2 Assumptions and compromises of the algorithm ensuring its universal applicability

Despite the strengths of our methodology, certain limitations must be acknowledged. The reliance on discrete differential geometries (Bobenko et al., 2008) means that the input dataset must represent a single, compact, and topologically connected structure (Thiele et al., 2016a). For objects separating from one to the next cross section into multiple strands, split algorithms are available. However, this comes at the cost of interpretability of the statistics of geometric properties. Moreover, the method functions optimally for convex hulls (Rockafellar, 1970), although a follow-up study will show, that the full geometric diversity of intrusive salt structures and crystalline bodies from the German subsurface can be quantitatively compared without major limitations. These assumptions should be considered when interpreting results in other domains.

The focus of this study was to establish a generalized algorithm to quantitatively describe the shape of objects and to infer dis-/similarity between geometries. Given the wide range of potential and available models, the algorithm requires some trade-offs to be universally applicable. Discussion of the data distributions for the geometrical parameters (see Fig. 6) focuses on the results from measuring the sphere, representing a comprehensible case with distinct expected data distributions: The nature of a sphere is a similar shape of any section through the center, eventually resulting in a normal distribution of the levelled distance measurements in both horizontal and vertical directions. This expected distribution is not produced in our case due to the generation approach of the orthogonal cross sections: The assembled sections follow the contour of the structure (see Fig. 5g & h), which results in larger measurements for the orthogonal horizontal data and a slightly tailed distribution of the combined horizontal data, similar to an ellipsoid with a low contrast in the main axes. As this situation is rarely seen in geological modeling, the impact is small since anisotropic geometries are measured accurately with our segmentation algorithm. The gradient and curvature data reflect the effects of our approach as well: While the gradient diagram of the sphere shows a symmetric distribution as expected, the relative elongation of the orthogonal sections increases the frequency of lower gradient measurements. Due to this accumulating effect, the presence of low-dipping surfaces of a structure is overestimated by the data. Furthermore, the exclusion of marginal cross sections leads to vertical clipping that introduces infinite gradient measurements (representing two consecutive vertices being exactly vertical) that would not exist when measuring the sections in a rounded, unclipped state. The curvature data is influenced by this clipping as well, that results in few large values where the three consecutive vertices form a large angle. These measurements increase the variance of curvature data considerably, with the majority of data for most datasets being located within the 0 to 5% and 5 to 10% bins.

However, the discriminability of the standard geometries and basic 3D objects in the cluster analysis is ensured despite these compromises made in the methodology: structures of varying anisotropy plot in different parts of cluster diagrams showing contribution of the combined horizontal data, as the highest contribution of horizontal data comes from the first and last distribution bins (see Fig. 10c). Similarly, as vertical and gradient data distributions of flattened geometries show the discussed characteristic properties, they differ in their PC scores from their rounded counterparts. Furthermore, the discussed increased frequency of gradient measurements around 0 does not change cluster patterns as it applies to all datasets. The same is true for the artifact-influenced curvature data and its impact on the general clustering of similar structures. Still, its

squeezed nature shows an effect on the overall clustering results, as the curvature data does not show any considerable contribution in PC dimensions.

540

### 4.3 Intended direct application and potential further usage of the quantification method and the standard geometries

The methodology will be a part of a larger framework to model and compare geological structures based on sparse data in the context of the German site selection. For most regions of interest for nuclear waste disposal, seismic 2D data are available frequently with a few boreholes. This is similar to the cross sections established through the benchmark models here, allowing for a fast model selection based on the geometrical properties and potentially further constraining hyper parameter selection for interpolation. However, for the integration of unmodeled sparse input data in the initial conceptualization of geological models, some adoptions of the workflow are needed. Obviously, the creation of segmented sections is omitted as the starting point are cross sections, which can be analyzed as described. The number of geometrical data is restricted by the number of available cross sections, thus a comparison will be conducted on a less complete statistical basis. Consequently, a user likely has to solely rely on the Kullback-Leibler divergence and cluster analysis to assess the reasonableness of various shapes. In case these analyses do not limit the range of standard geometries sufficiently, additional experience-based benchmarks could be created and clustered among the available models to test whether a closer fit applies. Here, more complex structural configurations could easily be approached by superposition of basic benchmarks. Thus, the choice of the conceptual model is based on the quantification and does not rely on the expert knowledge only. After interpolation resulted in a series of stochastic prior realizations, the method will be used for falsification by data (e.g. boreholes). Furthermore, the application of the framework to purely quantify the shape of a modeled 3D body can be very useful in the context of the site selection. Here, the comparative parts of our proposed analysis (i.e. Kullback-Leibler divergence and cluster analysis) might be of little value and thus be omitted. Additionally, applying our methodology also supports testing for the minimum amount of data necessary for geological modeling, as different data densities and configurations can be inserted into the algorithm. The open-access collection of benchmarks for geomodeling is also a convenient tool to visualize the range of three-dimensional geometries of the different rock types to a broader audience, which aids in the communication of uncertainties and decisions for geoscientists and stakeholders in various settings (see Zehner, 2021).

### 5 Conclusion

In our publication, we presented a methodology to quantitatively describe, compare and systematize 2D and 3D datasets, and proposed a set of regular standard geometries as benchmark models in geomodeling approaches. Demonstrating the quantification method on the 3D standard geometries, their geometrical dis-/similarity is assessed. The combined evaluation of data distributions and a cluster analysis reproduces the main geometrical characteristics of input meshes and visualizes differences between various datasets. While distributions of combined horizontal extensional measurements provide insight into

the anisotropy of datasets and the potential existence of overhangs, distributions of the vertical extent indicate the character of the top surface of structures and support or falsify the presence of overhangs. Distributions of gradient and curvature data (1) indicate the prevailing character of the slope of the lateral surfaces of structures, (2) further emphasize potentially present flat top surfaces and (3) give a general indicator on the sphericity of a structure. Cluster analysis of normalized, dimensionally reduced data groups and systematizes input structures based on the combined measured statistical parameters. In our application to synthetic datasets, clustering also serves to identify and exclude or merge benchmark models showing large geometrical similarity. Apart from cluster analysis and assessment of data distributions, comparison of parameter distributions is furthermore achieved using the Kullback-Leibler divergence. The proposed method and standard geometries are intended to be used at several stages within a workflow for structural geomodeling, both for initial conceptualization, potential adjustment of the interpolation method and examination of structural reasonableness of resulting models. Furthermore, general shape quantification for exploration/storage estimates can be realized.

As indicated earlier, the first follow-up study aims at applying the method to a large database of structural geological models. Afterwards, the method will be applied to datasets of sparse, unmodeled input data and coupled with a spatial interpolation algorithm in a study focusing on geomodeling based on progressively reduced datasets.

### Code and data availability

Method development was carried out in Python. The method mostly relies on the capabilities of the libraries Shapely (https://shapely.readthedocs.io/en/stable/), PyVista (https://pyvista.org/) and Plotly (https://plotly.com/). The python code, the condensed database of standard geometries (as .vtk-files) and the datasets of raw extensional, gradient and curvature data are stored at https://doi.org/10.5281/zenodo.15795851, (Carl, 2025).

### Author contribution

FC: writing–original draft; writing–review and editing; method development; data acquisition and analysis; conceptualization; visualization. JY: writing–review and editing. MCC: writing–review and editing. FW: writing–review and editing; supervision; funding acquisition. PA-Z: writing–review and editing; conceptualization; supervision; funding acquisition.

### Declarations

The authors declare the following financial interests/personal relationships which may be considered as potential competing interests: Friedrich Carl, Jian Yang and Marlise Colling Cassel are funded by the German Federal Company for Radioactive Waste Disposal (BGE). Other authors declare that they have no conflict of interest.

**Acknowledgements**

This study was conducted as part of the GeoBlocks project. GeoBlocks is a collaboration of RWTH Aachen University, University of Aberdeen and the German Federal Institute for Geosciences and Natural Resources (BGR) within the research cluster URS (Uncertainty and Robustness with regard to the Safety of a repository for high-level radioactive waste), funded by the German Federal Company for Radioactive Waste Disposal (BGE), with the main objective being the creation of an open-source workflow for geological modeling that includes the quantification and visualization of uncertainties and aims at the optimization of sampling procedures. We thank for the financial support by BGE. The authors would also like to thank BGR for the data provided to the study. Mark Lindsay and Mark Jessell are thanked for their constructive and thorough reviews, which helped to improve the paper significantly.

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

# Appendix

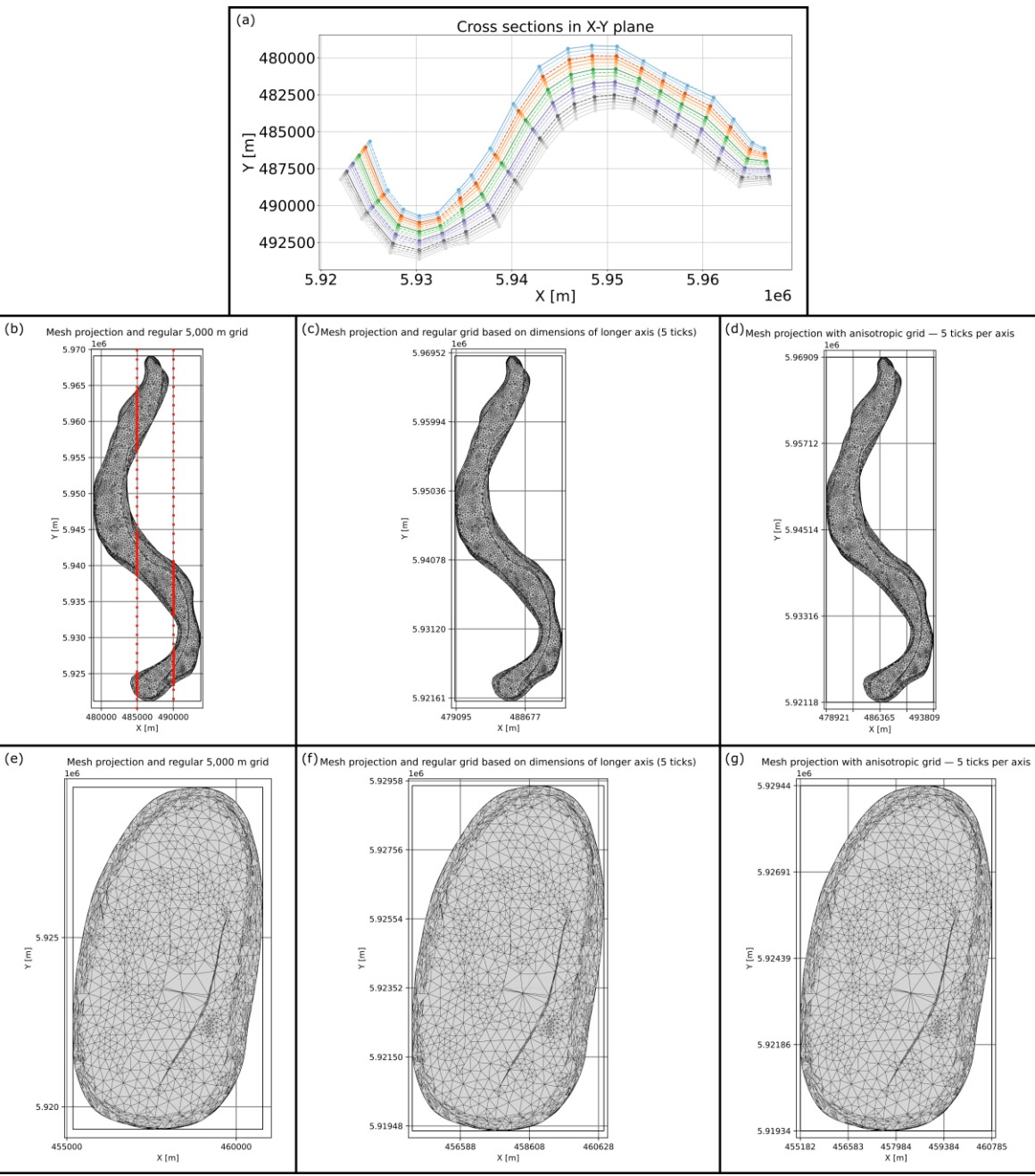

**Fig A1: Visualization of the advantages of measuring the extents of input meshes on sections normal to their horizontal main axes. Lower row: model "Seefeld" from BGR et al. (2022) for comparative purposes. a): Sections using the proposed method on "Altenbruch-Beverstedt" (top view). b) & e): hypothetical measurement of the horizontal extent along a regular grid (grey lines) of constant size for all datasets (example: 5000 m grid size for both). c) & f): measurement along a mesh-specific regular grid based on the extent of the longer axis of the mesh´s bounding box. d) & g): Measurement along an anisotropic grid to have an equal amount of sections per direction. Multiple cuts along a horizontal measuring line for an irregular structure are visualized in a)**

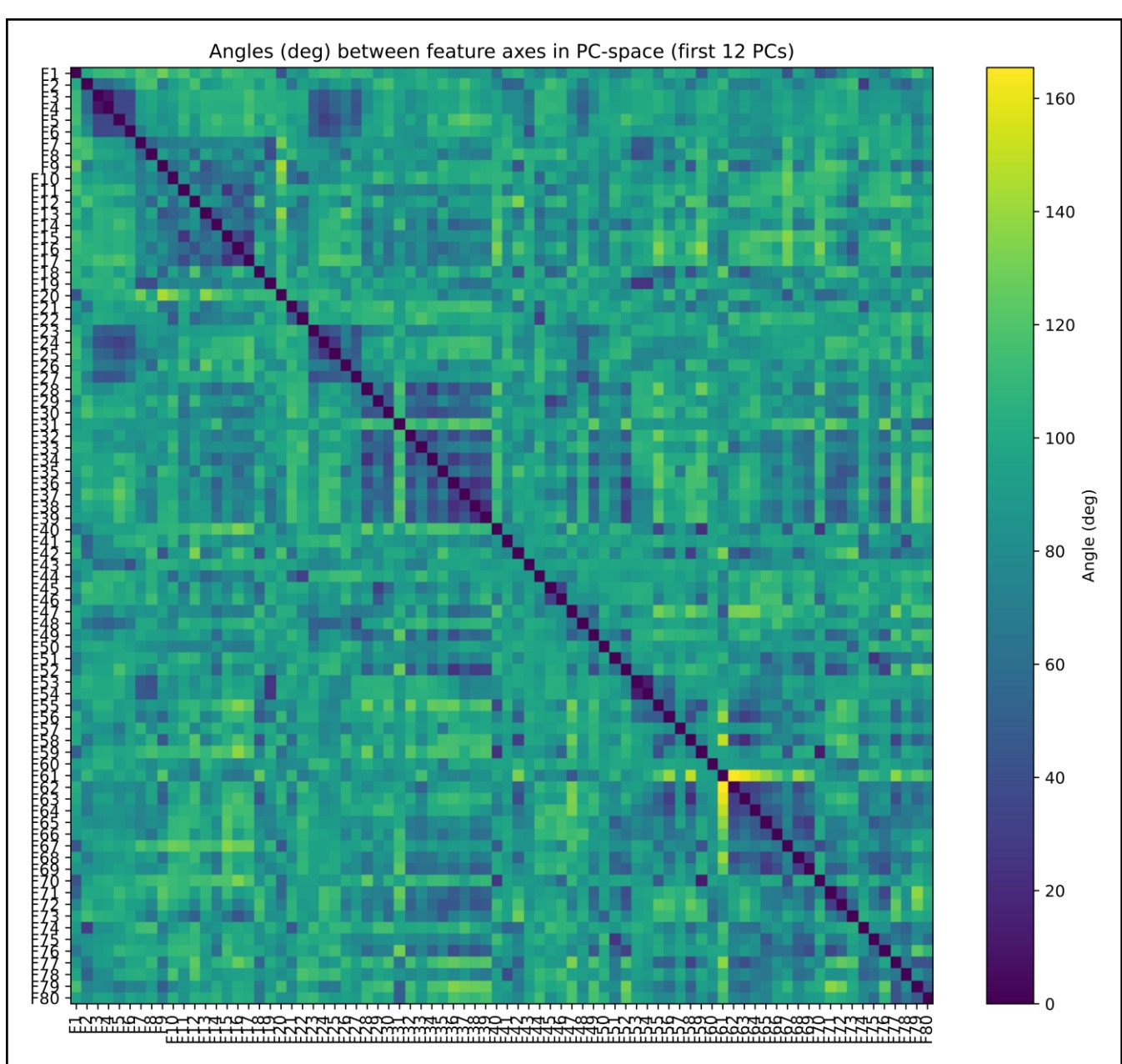

**Fig A2: Matrix plot for the angles between feature axes in PC space. The plot is used to assess the dependencies among the features (percentiles of the parameter PDF´s) in the cluster analysis. Small angles (dark blue) and large angles (bright yellowish) indicate strong dependency between individual features. This can be seen for instance between F3-F6; F32-F39 and F61-F69, with the exception of F67 (F61 is inversely dependant from F62-F64). The strong inter-feature dependencies result in a weak cluster separation beyond PC6. Thus, PC7 to 12 are not shown in Fig. 11**

700