# Peer review of "Quantitative comparison of three-dimensional bodies using geometrical properties to validate the dissimilarity of a standard collection of 3D geomodels"

_EGUsphere, 2025_

## Author Comment (AC1)

Report of the replies to the reviews and further unrelated changes

P 1-4: Reply to the review of Mark Lindsay

P 5-9: Reply to the review of Mark Jessell

P 9-10: Indication of changes not (directly) requested by the reviewers

Reply to the review of Mark Lindsay (RC1)

Dear Mark Lindsay,

We would like to thank you very much for the thorough review and the critical comments. Your constructive suggestions and corrections strongly improve the quality of our manuscript. On the following pages, we present our changes and corrections to your individual comments.

Kind regards,
Friedrich Carl and co-authors
* * *
A) The abstract will benefit from examples of application. Who is this aimed at? The second sentence of the introduction is a good example that would help here.

→ We agree that the abstract lacked a statement for the applicability of our method. Given the thorough edit of the introduction (see below), we decided to not directly name research fields but to outline the general usability of the method within a structural geomodeling workflow at the end of the abstract instead.
*"It is expected that the methodology and set of benchmark models will aid in advances to model, analyse and compare subsurface structures based on sparse data, as our framework can be used for an initial structural approximation prior to modeling, for the setup of the interpolation method and for the falsification of probabilistic model realizations after interpolation."*

B) I admit to being a bit lost in the first few paragraphs of the introduction. My understanding of "shape quantification" starts very simply, with volumes, surface areas, aspect ratios, elongation, flattening, and so on. 3D modelling software can do all this, so I wondered why things like cross sections, CNNs and Transformers were raised so early in the manuscript. I gathered that the intent is to quantify from multi-modal sources, such as sections, images, maps, etc. That is, not in a constructed 3D model itself. This is why you cite many sophisticated methods that are totally unnecessary if you have a 3D model, but are if they are static 2D representations of a 3D object (e.g. Multi-view approaches). Some 3D model-related studies are then cited (L61-65), so it remains confusing. I suggest opening the introduction with a clear description of shape quantification and the media through which it will be conducted. (I completely understand now I have read through to the method section. Please rewrite the introduction around a similar description).

→ We have understood that the introduction was confusing, both in its content and structure. We have tried to address these problems with a major revision of the introduction. As you suggested, Sect. 1.1 now starts with an overview of shape quantification (from the perspective of material science), fully focusing on "simple" shape parameters. The chapter then continues by shifting the focus to geoscientific studies and to a part reflecting on 1) the capabilities of 3D modelling software for shape quantification,

and 2) gaps in available software and methods to quantify lower-dimensional data and 3D objects direction-dependant. Then, we transition to pre-existing paragraphs

C) From what I can gather, the intent is to quantify the shape so it can be compared to a set of standard geometries, and then the modelled object can be given a label (e.g. wall(highly.anisotropic_hourglass-shape_rounded)). Once you know the label, then you can make some interpretation of the geological history? Usually when you build a model, the geologist has a good idea (i.e. conceptual model) about what the intended object should look like. Obvisouly this has subjective bias behind it. Thus, what you method does is to check whether the desired object is close to what it should look like. If the object respects the data, but not the conceptual model, then it could more questions about what the geological history could be. If this is what you are doing this for, that's great, and good science. Please make that clear in the abstract and introduction. (I didn't come to that realisation until the results and discussion)

→ We acknowledge that the first manuscript version did not describe clearly, what the intended application of our method and benchmark models is within a geomodeling workflow. We have dedicated a paragraph to this question towards the end of the revised Sect. 1.3

D) Carl et al 2023 is cited and useful to read (and view, as there is a video), however is not available from RING without the necessary login credentials. You need to summarise this paper given it describes more fully the concept of "standard geometries" (at least in this context). Admittedly I only found one version, and may have missed any open-access.

→ We have clarified in Sect. 1.1, that we actually already summarized/reviewed the relevant part of the conference paper in our first manuscript version. Furthermore, we have successfully requested, that the conference paper is openly accessible, under: https://www.ring-team.org/research-publications/ring-meeting-papers?view=pub&id=223007
*"In what follows, the concept of "standard geometries" initially described by Carl et al. (2023) as a geometrical systematization to collect and catalogue subsurface geometries of the potential host rocks in the German site selection for a nuclear waste repository (halite rock, claystone and crystalline rocks) is reviewed, adapted and extended."*

E) The introduction to topology with conformable, unconformable, concordant and discordant is good, however it's not clear why you have introduced it with respect to shapes. If you are making a point about how geological history -> topology -> modelled shape and that can be quantified, that's great, but you need to be quite explicit about that.

→ We see that the usage of these terms can cause confusion. As the geometrical systematization of salt and the different groups of crystalline rocks is based on these topological terms and basics in the literature, we decided to adapt these terms to claystones as well, although it had not been directly established like this in the literature we used. The overall reason for explaining the setup of the geometrical systematization was to introduce the concept of the standard geometries, whose dis-/similarity is assessed and quantified in the results. Thus, we do not claim (yet) that we can infer the geological history of a structure from our method results. However, the context between the method results and regional geology is in fact one of the research questions assessed in a follow-up study that was recently submitted to the *Journal of Structural Geology*.

We have added the following sentence for clarification: *"Please note that the classification is purely geometric, even though the terminology of subdividing categories can also be found in topological considerations (see for instance Thiele et al., 2016a, b)"*

F) This section, especially the description of various halite geometries would benefit from reference to figure 2. It's quite hard to follow without a visual representation, and geologists like pictures. Also, some field photos would be nice, but not critical.

→ We implemented references to Fig. 2 throughout this chapter as requested. We have decided to not include field photos since we already utilize several large figures in the introduction. Instead, we have referred to literature where photos can be viewed.

G) What are the four thin and unlabelled objects at the end of figure 2?

→ We would like to point out that the labels of the objects in Fig. 2 are above the geometries. However, we acknowledge that these four structures have been too small, so that their differences were hard to identify. Thus, we increased their size.

H) The approach needs a figure showing the entire workflow from the initial vtk to the final computation of gradients and curvatures. You could add the figure to the pseudo code in Fig 3 by running down alongside it. (reading on) an improved version of figure 4 would do (see additional comments below).

Fig. 4.: The text is too small, and quality of the images not adequate for publication. Screen shots are okay sometimes, but if there is text, one needs to be able to read it (e.g. the coordinates, the key, legends etc)

→ We fully recognize that the text and certain other elements in the Fig. 5 (formerly Fig. 4) were too small, and that image quality was partially lacking. We increased the font sizes and tried to address the quality issues.

I) You are comparing a generic sphere with a model of Altenbruch-Beverstedt. So you need a section in the introduction describing Altenbruch-Beverstedt otherwise we have no point of reference to know whether your results are meaningful given the structure we would expect to be quantified. You also need an image of the geological model, or the structure you have picked out from it for the analysis.

→ We have implemented a new Section (Sect. 1.2) that addresses your request: the model is shown, its geological background is described shortly and we state, why this structure was chosen as a representative example. The first paragraph of the results-section was also adapted accordingly.

J) PCA – I'd be careful about interpreting too much from anything beyond PC6. You have stopped there, but the number of PCs to get to 90% indicates a pretty complex and high-dimensional data set. Two things:

Check if your metrics are dependent. Use an SPLOM to find out and include in the manuscript. Strong dependency can explain why need 12 components.

I wouldn't bother showing anything beyond a comparison of PC5 and 6 in Figure 9. You don't present these in the results.

→ We have thankfully followed your suggestion and created a matrix plot for the angles between the feature axes in PC-space, visible in the appendix. As you suspected, there are several groups of adjacent features showing stronger dependencies. We have adjusted the cluster plot (formerly Fig. 9, now Fig. 11) accordingly and also added a short explanation at the end of Sect. 2.2 and the beginning of Sect. 3.3.

*"A feature angle matrix (Fig. A2) was computed to check the dependencies between the percentiles of the PDF´s. As a strong dependency was identified within several groups of features, the principal component matrix plot (Fig. 11) was limited to the first six PC´s."*

K) L83 Are you introducing grain size as you'll be quantifying their shapes? It doesn't seem to have much to do with the rest of the paragraph.

→ We agree that this information is unnecessary, we have adapted this part in Sect. 1.1 accordingly (also according to a suggestion of Mark Jessell):

*"Claystones and shales are clastic sedimentary rocks which are commonly deposited conformably onto the underlying strata (Selley, 2000; see also Fig. 1, upper section)."*

L) L86 "tilting and folding *of flat-lying structure* can result in a range of geometries that remain generally conformable."

→ We have added your suggestion

M) L100 "Crystalline rocks considered are both plutonic and high-grade metamorphic rocks (migmatites and gneisses)." Crystalline rock is a catch-all term meaning the basement to an overlying sedimentary basin. Technically, high-grade plutonic rocks end up as gneisses, but are not "both". Also crystalline rock can be extrusive, low and medium grade metamorphic, just depends on where you are.

→ We see that our wording had been ambiguous here. We wanted to say, that the crystalline rocks considered in the German site selection (and thus in our geometrical systematization) are plutonic rocks as well as high-grade metamorphic rocks (migmatites and gneisses). This combination had been defined by the German federal company for radioactive waste disposal (BGE). Thus, we have adapted the sentence as follows:

*"Crystalline rocks considered in the context of the German site selection are plutonic rocks as well as high-grade metamorphic rocks (migmatites and gneisses)"*

N) L146 What kind of clustering? Eg. KNN? Or something else

→ We have added the clarification: K-means clustering

O) L149 "Our method cannot be used to quantitatively compare implicit representations of structures." so not directly from a scalar field (e.g. Geomodeller or Leapfrog) – if so I'd be clear about that, because it reads like you can't use your method on any object rendered from an implicit method, while I'm sure you can!

→ We have adjusted the sentence to *"Our method cannot be used to quantitatively compare implicit representations of structures directly from a scalar field, though."*. It was moved slightly downwards in Sect. 1.3

P) L216 Assuming these moments are centred around the mean?

→ Your comment made us realize thankfully, that we had made a mistake here, as we decided in the end of the initial method creation to only use three statistical moments (mean, variance, skewness; all naturally centred around the mean) plus the median and standard deviation. We corrected the text in Sect 2.2 & 3.2 and Table 1 accordingly.

Dear Mark Jessell,

We would like to thank you very much for the thorough review and your suggested changes. Your input strongly improves the quality of our manuscript. On the following pages, we present our changes and corrections to your individual comments.

Kind regards,
Friedrich Carl and co-authors
* * *
A) Line 33: The introduction begins with the observation: "Accurate shape quantification independent of the objects' orientation is essential for applications such as geological modelling, resource management and structural analysis, where understanding the geometric properties of objects can inform decision making and enhance predictive capabilities"

This is key, as the choice of model comparison method fundamentally depends on the use to which the models are to be put. A model for constraining a geophysical inversion has to retain distinct petrophysical domains, but could merge others with similar properties. A model to calculate transport properties (fluid flow, electrical conductivity... is concerned both with rock physics and adjacent body topology. A structural interpretation needs to understand age relationships, not just geometry.

It is not at all clear to me what specific scientific question is being addressed by this method, that is limited to the study of isolated convex hulls with no branching structures? Why do we want to compare two models of bodies around salt domes, when with the 3D seismic we actually have a detailed description of the geometry of the object? What do we learn from comparing it with another potentially similar object? All this needs to be addressed to underpin this work. I suggest that the paper by Parquer et al 2025 (https://gmd.copernicus.org/articles/18/71/2025/) will provide some interesting insights into this topic.

→ We recognize that the structure of the first manuscript was lacking and that we did not clarify enough what the method and benchmark models are supposed to offer. We thoroughly revised the introduction to address all comments related to that section. The point at the beginning of Sect. 4.2 regarding the convex hulls has been clarified referring to a follow-up study. The intended application of our method and benchmark models within a geomodeling workflow is mainly addressed towards the end of the revised Sect. 1.3, although the build-up to this question starts in Sect. 1.1. The paper by Parquer et al. (2025) has been carefully studied and incorporated in the introduction.

B) Section 4.3  Given this analysis, it seems to me this should be a part 2 of the methods and results, and not placed in the discussion as we are essentially revisiting the entire methodology. I suggest that the authors introduce the full and restricted model sets from the beginning, compare the methods and results and then discuss this, rather than the structure of the ms currently used.

→ We have largely followed your suggestion and redistributed most of Sect. 4.3 and the accompanying Fig. (formerly Fig. 10) to different parts of Sect. 3.2 and 3.3. We have only decided to keep a short paragraph of Sect 4.3 in the discussion and moved it to Sect 4.2, since its content is directly related to that section. Furthermore, the content of the former Sect. 4.4 ("Online collection of benchmark geometries") was moved to Sect. 3.3 as well (and changed a bit in its wording), as it had been directly related to the moved content of Sect. 4.3

C) Line 274: If we have a method that, when applied to the simplest possible shape (a sphere) tells us "...the most similar models regarding the respective distributions of the six parameters are the "sheet(cylindric_rounded)" for the horizontal data of the first direction, the "prism" for the orthogonal horizontal data, the "batholithV3" for the combined horizontal data, the "anticline_wall (rounded)_batholithV1" for the vertical data, the "phacolith" model for the gradients and the "ellipsoid" for the curvatures".

Is this really a selling point for the method? It sounds a bit like the apocryphal sightless committees' description of an elephant. It raises the question as to why a sphere is not the best descriptor of a sphere, and if not is this really a useful method?

→ We believe that there is a misunderstanding here. When using the Kullback-Leibler divergence in order to determine which regular structure is most similar to a test structure (here: the sphere) for a given parameter, the test structure is compared to all other structures of the benchmark set except itself. Computing the Kullback Leibler divergence of a distribution with itself results in a value of 0 (=similarity), which would not have any informational value in our case. Given that the overall method is designed to compare a random irregular structure to other irregular bodies or regular benchmarks, computing the Kullback-Leibler divergence serves the purpose of identifying similar structures regarding a given parameter.

Nevertheless, we understand now that this paragraph might cause confusion and have added a clarification: "*The similarity of two distributions is larger, the smaller the KL divergence is, with a value of 0 indicating equality of the distributions (obtained for instance when comparing a structure with itself).*"

D) Line 84: "Since all clastic sediments are initially deposited conformably onto the underlying strata" not sure why this has to be true, can't clastic sediments deposit on erosional surfaces?

→ We agree with you. In accordance with a suggestion from Mark Lindsay, we adapted the sentence to *"Claystones and shales are clastic sedimentary rocks which are commonly deposited conformably onto the underlying strata (Selley, 2000; see also Fig. 1, upper section)"*

E) Line 104 "Their current shape depends not only on the geometry of the original rock body but also on the specific mineral assemblage of the protolith" Why does the currently geometry depend on the mineral assemblage? Similarly, why does it depend on the "the pressure-temperature conditions experienced during metamorphism"

→ We agree. We have adapted the sentence to *"Their current shape depends not only on the geometry of the original rock body but also on the specific deformation history experienced during metamorphism"*

F) Line 106 "Overall, most high-grade metamorphic rock bodies in the German subsurface are bounded by either plutonic intrusions or fault zones" Presumably the shallower ones, of interest to this study, are mostly bound on their top-most surface by unconformities?

→ We were initially referring only to the lateral bounds. As your valuable comment points out that we did not touch on the vertical bounds, we adapted the text to: *"Overall, most high-grade metamorphic rock bodies in the German subsurface are laterally bounded by either plutonic intrusions or fault zones and their top is either bound by unconformities or represents the present-day topography in most cases"*

G) Figure 1. This is actually a part of the methodology, not the introductory material in my opinion and should appear there, together with the description.

→ We realized that we had not made clear properly, that this part and the figure had been largely established earlier (in Carl et al., 2023). Thus, we believe that it should not be part of the methodology but stay in the introduction, as it is largely a review of a previous study.

H) Figure 1. Not sure if lateral stratigraphic pinching out of horizons is accounted for in this scheme, but occurs in Fig 2 so maybe I am missing something?

→ We agree that we missed that. We corrected it in the text and in the Figs. 1, 2 and 12 *("Lateral stratigraphic pinchout is conformable proximally but results in an unconformity at its tip")*

I) Figure 2. Some of the shapes (e.g. anticline_wall) are difficult to understand in terms of their 3D geometry as much of the body is hidden, perhaps consider semitransparent front halves for some of these models?

→ We agree. After testing several models in question, we adjusted the transparency to the "anticline_wall(hourglass-shape_flattened", "anticline_wall(hourglass-shape_rounded" and "roller" in Figs. 2 and 12

J) Figure 2. What about more complex fold geometries such as refolded folds, or even simple saddles?

→ As we established the geometrical systematization for the German site selection, where most where most structurally complex bodies have already been excluded from the considerations, we did not include more complex fold geometries. However, we addressed this topic in the revised Sect. 4.3: *"In case these analyses do not limit the range of standard geometries sufficiently, additional experience-based benchmarks could be created and clustered among the available models to test whether a closer fit applies. Here, more complex structural configurations could easily be approached by superposition of basic benchmarks [...]"*

K) Line 167 "the input file format has to be changed if it is not .vtk" Maybe just say: "The system only supports .vtk file formats".

→We decided to write *"The system currently only supports .vtk file formats"*

L) Line 170 "which first rasters" should be "which first discretises"

→ We changed the line according to your suggestion.

M) Line 173: "Orientation of sections normal to the longitudinal axis of the structure (first direction)" What if the structure doesn't have a simple geometry that you can assign a longitudinal axis to? Does this system allow bounding boxes that do not have x & y axes parallel to real-world horizonal planes?

→ We see that our description has been misleading. We have clarified that the sections of the first set are rotated individually, with the rotation step showing the minimal area respectively being oriented normal to the longitudinal axis (compare also Fig. 4 & 5h): *"Orientation of sections normal to the*

*longitudinal axis of the structure (first direction) have been determined by a minimization of the cross-sectional area, as sections are sequentially rotated (Stephenson, 2018; Fig. 4, Part 1)."*

N) Line 176" "Extensional measurements are conducted on each cross section at 5 equidistant transects" Why 5, did you perform studies to show this was optimal?

→ We did not perform quantitative studies on the optimal amount of sections to be created and the amount of extensional measurements per section, since this would have been necessary for every potential input structure individually. This was explicitly not what we wanted, since we aimed at creating a scale-independent method.

The exact number of 5 measurements per 20 sections for all datasets was chosen after some trials, as in the cluster analysis, the corresponding PDF´s were actually defined by 200 data points per used extensional parameter that way (100 measurements per horizontal extent = 200 points for the utilized combined horizontal data; 5 vertical measurements for in total 40 sections = 200 vertical data points).

The amount was visually found to resolve the distributional shape of differently-sized irregular structures sufficiently well (this is addressed shortly in the mentioned follow-up study). As cluster differentiation is based on these data distributions (plus gradient and curvature distributions), we assumed that more data would also not affect cluster results considerably. Meanwhile, processing times were ensured to remain reasonable: When applying the methodology to a very irregular structure, artifact corrections can get time-consuming (potentially longer, the more sections are computed). For a few structures (irregular ones covered in the follow-up study), with 40 sections overall, it already took several hours.

O) Line 177: "Since the very first and last cross section of both directions are excluded from the measurements as they would (undesirably) slice irregular polygons several times" I don't follow this logic, if the body is really irregular any section may intersect it multiple times?

→ We believe that your confusion is based on our misleading wording in (formerly) line 173 (see above: comment M), as we had not made clear enough how/why the sections of both directions are indeed oriented normal to the two horizontal main axes (see Fig. 5h), thus avoiding multiple intersections in irregular structures (see also Fig. A1). In case we misunderstood your comment, we would kindly ask you to clarify your point.

P) Line 179: "20 intervals are considered for every input structure" Why 20, did you perform studies to show this was optimal?

→ please refer to comment N)

Q) Line 191: The assertions in this paragraph are hard to understand, maybe it needs a figure (even in sup materials would do?)

→ We have created Fig. A1 in the appendix to address the confusion and added references in this paragraph. We hope, that this Figure can clarify sources of confusion you addressed in the previous comments M) & O)

R) Section 3.3 This section would benefit from some sub-structure as it is hard to digest for the uninitiated.

→ We acknowledge that the structure has been suboptimal, so we tried to improve the text by ensuring that for every principal component, information is provided in the following order:

-contributions to PC
-effects on cluster separation
-geometric cluster interpretation

We furthermore arranged the text into one paragraph per PC-PC cluster plot in Fig. 11 (formerly Fig. 9), and adapted the first sentence of every of these paragraphs.

S) Fig. 4c, d, f, g, h, i, j text way too small to read.

→ We see that the text and certain other elements in the Fig. 5 (formerly Fig. 4) were too small, and that image quality was partially lacking. We increased the font sizes and tried to address the quality issues.
* * *
Changes not (directly) requested by the reviewers:

- after internal discussions, we have decided to change the order of the author list (the involved people stay the same)

- the order of figures has partially been changed, reflecting the thorough revision of the introduction and Sect. 4.3. Some figure captions have been adapted as well, due to the same reasons and in accordance with other comments. Changes to the order of existing figures are the following:

New Fig 3; Fig. 4 → Fig. 5; Fig. 5 → Fig. 6; Fig 6→ Fig. 7; Fig. 10 → Fig 8 (moved to Sect. 3.1); Fig. 7 → Fig. 9; Fig 8 → Fig 10; Fig. 9 → Fig 11; Fig 11 → Fig 12; new Figs. A1 & A2

- Throughout the manuscript, we added clarifications at several points that the method can also be applied to 2D data

- the first paragraph of Sect. 3 has been changed as some information is now already included in the new Sect. 1.2

- the first paragraph of Sect. 4.1 has been revised according to the changes made to the introduction

-small changes have been made to Sect. 4.2: a change to the first paragraph is already indicated above. Then, there is the inclusion of *"[...] quantitatively describe the shape of objects and to [...]"* in the first sentence of the second paragraph. At the end of Sect. 4.2 (formerly belonging to Sect. 4.3), potentially misleading wording (*"However, the discriminability of the standard geometries and basic 3D objects in the cluster analysis"*) are clarified and a mistake made by the authors in the first manuscript version (*"Similarly,  vertical and gradient data distributions"*) is corrected.

-Sect. 4.3(formerly 4.5): The title has been changed to *" Intended direct application and potential further usage of the quantification method and the standard geometries"*. The section has been expanded to address the overall changes implemented mainly in the introduction

-Conclusion: the following sentence has been added at the end of the first paragraph: "*The proposed method and benchmark models are intended to be used at several stages within a workflow for structural geomodeling, both for initial conceptualization, potential adjustment of the interpolation method and examination of structural reasonableness of resulting models. Furthermore, general shape quantification for exploration/storage estimates can be realized.*"

-Conclusion: the last sentence has been adapted to better reflect the upcoming plans of the project, as they have become more concrete since initially submitting our manuscript. *"Afterwards, the method will be applied to datasets of sparse, unmodeled input data and coupled with a spatial interpolation algorithm in a study focusing on geomodeling based on progressively reduced datasets."*

-some headings had not been formatted correctly (heading type 3 instead of type 2) → this should be corrected now

-the reference list should be formatted correctly now

---

## Author Response (AR2)

Report of the minor revisions requested for figures

Dear Jacqueline Reber, Mark Lindsay and Mark Jessell,

We would like to thank you very much for your suggestions to improve the readability of several figures. The following changes have been implemented as requested:

- The axes have been removed for Figs. 2 and 12. This allowed for a slight reduction of the height of both figures as well.

- Font size has been increased in Fig. 4. This required some minor changes to the layout/size of the individual pictures (especially d) and e)).

- Font size has been increased in Fig. 8. This required the position of the numbering within diagrams to shift from the upper left to the lower left, and some other minimal layout changes.

- Font size has been increased in Fig. 9. As the size of the individual legends increased accordingly, its respective position has been adapted for some diagrams to avoid overlap between legend and data bins as much as possible.

-Additionally, we corrected a typo in the author affiliations

Kind regards,
Friedrich Carl and co-authors

---

## Author Response (AR3)

Report of minor revisions

Dear Editors and Reviewers,

We would like to point out the minor changes made during the latest round of revisions:

- we have added the word "often" in line 42.

- some minor improvements were done to the relative positioning of some figure elements without changing the figure contents (minor shifting of figure elements by few millimeters at max to improve alignment of figure elements).

- some duplicate spaces were deleted.

- the order of Sections at the end of the manuscript was adjusted to follow the manuscript composition as specified on the respective website: the appendices have been placed directly after the conclusions. This resulted in the manuscript being 35 pages long instead of 34.

Kind regards,
Friedrich Carl and co-authors